# Polygenic adaptation from standing genetic variation allows rapid ecotype formation

**Nico Fuhrmann, Celine Prakash, Tobias S Kaiser***

Max Planck Institute for Evolutionary Biology, Plön, Germany

**Abstract** Adaptive ecotype formation can be the first step to speciation, but the genetic underpinnings of this process are poorly understood. Marine midges of the genus *Clunio* (Diptera) have recolonized Northern European shore areas after the last glaciation. In response to local tide conditions they have formed different ecotypes with respect to timing of adult emergence, oviposition behavior and larval habitat. Genomic analysis confirms the recent establishment of these ecotypes, reflected in massive haplotype sharing between ecotypes, irrespective of whether there is ongoing gene flow or geographic isolation. QTL mapping and genome screens reveal patterns of polygenic adaptation from standing genetic variation. Ecotype-associated loci prominently include circadian clock genes, as well as genes affecting sensory perception and nervous system development, hinting to a central role of these processes in ecotype formation. Our data show that adaptive ecotype formation can occur rapidly, with ongoing gene flow and largely based on a re-assortment of existing alleles.

## Editor's evaluation

This valuable study combines phenotypic analysis, quantitative genetics and population genomics to provide solid evidence for multiple genes underlying adaptive divergence in a marine midge system linked to tidal rhythm. Genes with a plausible role in perceiving and responding to lunar information are among the loci that most highly differentiate populations with distinct behaviors.

*For correspondence:
kaiser@evolbio.mpg.de

**Competing interest:** The authors declare that no competing interests exist.

## Introduction

The genetic foundations for rapid ecological adaptation and speciation remain poorly understood. Decades of research have focused on identifying single or few major genetic loci in such contexts, with some notable successes (*Wu and Ting, 2004*; *Phadnis and Orr, 2009*; *Nosil and Schluter, 2011*; *Van't Hof et al., 2016*). However, many genome-wide association studies have now shown that quantitative trait phenotypes have usually a polygenic basis (*Sella and Barton, 2019*; *Boyle et al., 2017*). This implies that also rapid natural adaptation should be expected to occur in a polygenic context, based on subtle allele frequency changes at many loci with small effects (*Barton, 2022*; *Barghi et al., 2020*; *Le Corre and Kremer, 2012*). A classic example for polygenic adaptation is human height (*Turchin et al., 2012*; *Robinson et al., 2015*), and other examples from humans may include birth weight and female hip size (*Field et al., 2016*). However, powerful datasets to detect polygenic adaptations are still mostly limited to humans and agriculturally relevant species (*Barghi et al., 2020*), and examples of polygenic adaptation in natural populations are rare. To show that polygenic adaptation occurs in natural populations requires an exceptional biogeographic setting, as well as deep genomic and genetic analysis. One such example may be found in *Midas* cichlids, where a polygenic trait architecture was found to promote rapid and stable sympatric speciation (*Kautt et al., 2020*). This finding

is particularly interesting, since theoretical work suggested that under scenarios with gene flow the genetic architecture of adaptation may – compared to adaptation without gene flow – rather consist of fewer, tightly linked loci with larger effects (*Yeaman and Whitlock, 2011*). Notably, the same study also suggested that the resulting large-effect QTL may often be composed of many tightly linked loci of smaller effects (*Yeaman and Whitlock, 2011*). Without migration we expect an exponential distribution of allele effects, with few large effect loci and many loci of small effects (*Barghi et al., 2020*; *Orr, 1998*).

In this study, we examine local adaptation and population divergence in another exceptional biogeographic setting, namely the postglacial evolution of new ecotypes in marine midges of the genus *Clunio* (Diptera: Chironomidae). In adaptation to their habitat, *Clunio* midges differ in oviposition behavior and reproductive timing, involving both circadian and circalunar clocks.

Circalunar clocks are biological time-keeping mechanisms that allow organisms to anticipate lunar phase (*Neumann, 2014*). Their molecular basis is unknown (*Andreatta and Tessmar-Raible, 2020*), making identification of adaptive loci for lunar timing both particularly challenging and interesting. In many marine organisms, circalunar clocks synchronize reproduction within a population. In *Clunio marinus* they have additional ecological relevance (*Kaiser, 2014*). Living in the intertidal zone of the European Atlantic coasts, *C. marinus* requires the habitat to be exposed by the tide for successful oviposition. The habitat is maximally exposed during the low waters of spring tide days around full moon and new moon. Adult emergence is restricted to these occasions by a circalunar clock, which tightly regulates development and maturation. Additionally, a circadian clock ensures emergence during only one of the two daily low tides. The adults reproduce immediately after emergence and die few hours later. As tidal regimes vary dramatically along the coastline, populations of this *Atlantic ecotype* of *C. marinus* show local timing adaptations (*Kaiser, 2014*; *Neumann, 1967*; *Kaiser et al., 2011*) for which the genetic underpinnings are partially known (*Kaiser et al., 2016*; *Kaiser and Heckel, 2012*).

When *Clunio* expanded into the North after the regression of the ice shield, new ecotypes evolved in the Baltic Sea (*Remmert, 1955*; *Palmén and Lindeberg, 1959*) and in the high Arctic (*Neumann and Honegger, 1969*; *Pflüger and Neumann, 1971*), which differ from the *Atlantic ecotype* of *C. marinus* in various ways (see *Figure 1* for a summary of defining characteristics of the three ecotypes). In the Baltic Sea the tides are negligible and the *Baltic ecotype* oviposits on the open water, from where the eggs quickly sink to the submerged larval habitat at water depths of up to 20 m (*Remmert, 1955*; *Endraß, 1976a*). Reproduction of the *Baltic ecotype* happens every day precisely at dusk under control of a circadian clock (*Heimbach, 1978*), without a detectable circalunar rhythm (*Endraß, 1976a*). Near Bergen (Norway) the *Baltic* and *Atlantic ecotypes* were reported to co-occur in sympatry, but in temporal reproductive isolation. The *Baltic ecotype* reproduces at dusk, the *Atlantic ecotype* reproduces during the afternoon low tide (*Heimbach, 1978*). Therefore, the *Baltic ecotype* is currently considered a separate species – *C. balticus*. However, *C. balticus* and *C. marinus* can be successfully interbred in the laboratory (*Heimbach, 1978*).

In the high Arctic there are normal tides and the *Arctic ecotype* of *C. marinus* is found in intertidal habitats (*Neumann and Honegger, 1969*). During its reproductive season, the permanent light of polar day precludes synchronization of the circadian and circalunar clocks with the environment. Thus, the *Arctic ecotype* relies on a so-called tidal hourglass timer, which allows it to emerge and reproduce during every low tide (*Pflüger and Neumann, 1971*). It does not show circalunar or circadian rhythms (*Pflüger and Neumann, 1971*).

The geological history of Northern Europe (*Patton et al., 2017*) implies the Baltic Sea and the high Arctic could only be colonized by *Clunio* after the last glacial ice shield regressed about 10,000 years ago. This inference is also supported by subfossil *Clunio* head capsules in Baltic Sea sediment cores (*Hofmann and Winn, 2000*). The new adaptations to the absence of tides in the Baltic Sea and to polar day in the high Arctic must therefore have occurred within this time frame. One strength of our model system is that we know the major adaptations specific to each ecotype, that is we know the major axes of recent selection. This makes *Clunio* a particularly interesting model system for studying the genetics of rapid adaptation. At the same time it promises to yield insights into the genetic pathways underlying the ecotype characteristics, including circadian and circalunar timing.

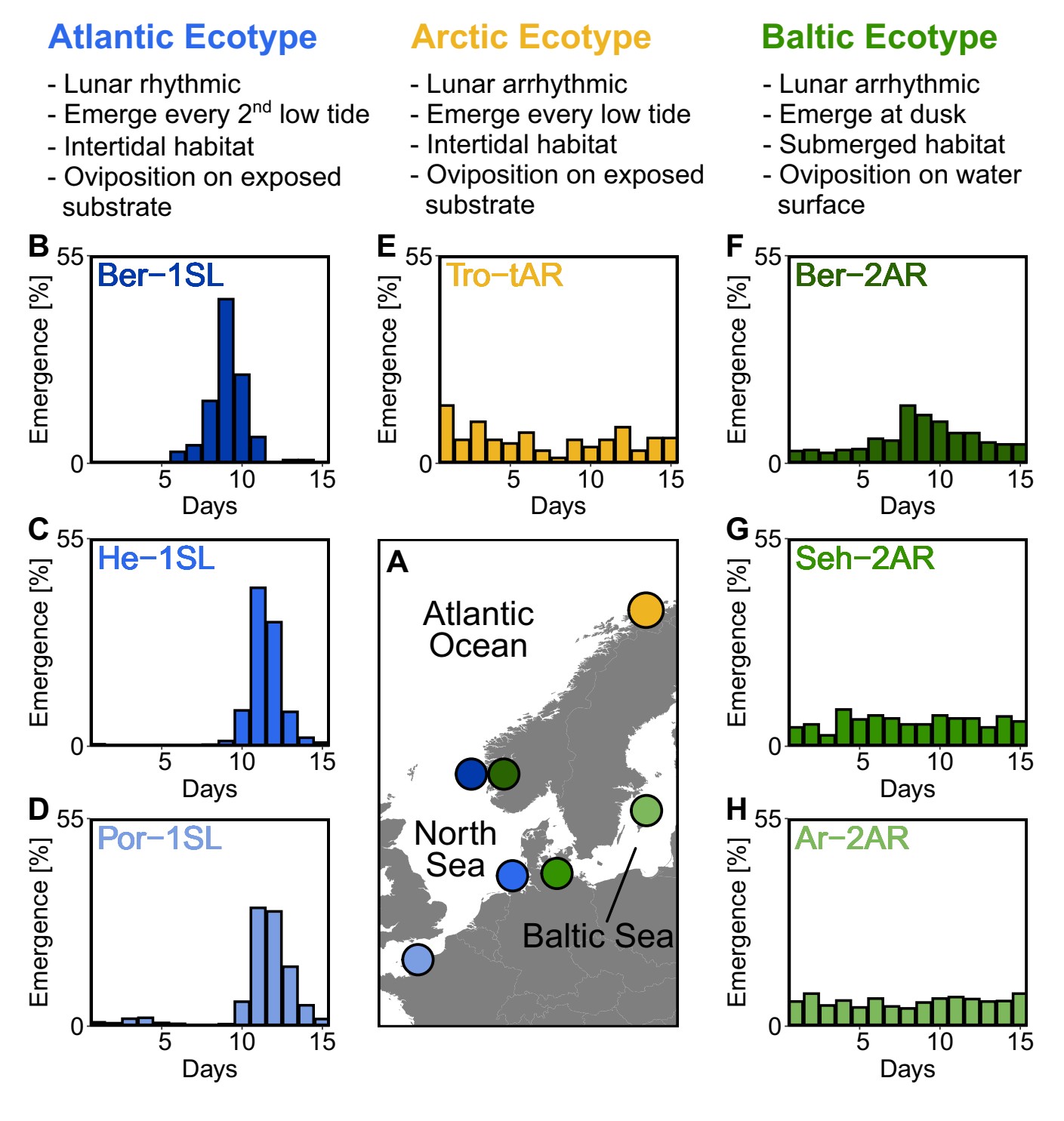

**Figure 1.** Northern European ecotypes of *Clunio* and their circalunar rhythms. The *Atlantic, Arctic* and *Baltic ecotypes* of *Clunio* differ mainly in their circalunar rhythms (**B–H**), circadian rhythms (*Figure 1—figure supplement 1*), as well as their habitat and the resulting oviposition behavior (see 'Assessment of oviposition behavior' in Methods section). (**A**) Sampling sites for this study. (**B–H**) Circalunar rhythms of adult emergence in corresponding laboratory strains under common garden conditions, with 16 hours of daylight and simulated tidal turbulence cycles to synchronize the lunar rhythm. In *Arctic* and *Baltic ecotypes* the lunar rhythm is absent (**E,G,H**) or very weak (**F**). Por-1SL: n=1,263; He-1SL: n=2,075; Ber-1SL: n=230; Tro-tAR: n=209; Ber-2AR: n=399; Seh-2AR: n=380; Ar-2AR: n=765.

The online version of this article includes the following figure supplement(s) for figure 1:

**Figure supplement 1.** Circadian emergence rhythm of the studied *Clunio* strains under laboratory conditions.

## Results

### *Clunio* ecotypes

Starting from field work in Northern Europe (*Figure 1A*), we established one laboratory strain of the *Arctic ecotype* from Tromsø (Norway, Tro-tAR; see methods for strain nomenclature) and three laboratory strains of the *Baltic ecotype*, from Bergen (Norway, Ber-2AR), Sehlendorf (Germany; Seh-2AR) and Ar (Sweden; Ar-2AR). We also established a strain of the *Atlantic ecotype* from Bergen (Ber-1SL, sympatric with Ber-2AR) and used two existing *Atlantic ecotype* laboratory strains from Helgoland (Germany; He-1SL) and Port-en-Bessin (France; Por-1SL). We confirmed the identity of the ecotypes in the laboratory by the absence of a lunar rhythm in the *Baltic* and *Arctic ecotypes* (*Figure 1B–H*), their circadian rhythm (*Figure 1—figure supplement 1B–H*) and their oviposition behavior (for details see methods section). The *Baltic ecotype* from Bergen (Ber-2AR, *Figure 1F*) was found weakly lunar-rhythmic.

### Evolutionary history and species status

We sequenced the full nuclear and mitochondrial genomes of 168 field-caught individuals from six geographic sites, including the two sympatric population ecotypes from Bergen (*Figure 1*). Based on a set of 792,032 single nucleotide polymorphisms (SNPs), we first investigated population structure and evolutionary history by performing a principal component analysis (PCA; *Figure 2A–B*) and testing for genetic admixture (*Figure 2C*). We also constructed a haplotype network of complete mitochondrial genomes (*Figure 2D*). There are several major observations that can be derived from these data.

First, there is strong geographic isolation between populations from different sites. In PCA, clusters are formed according to geography (*Figure 2A–B*, *Figure 2—figure supplement 1*). Mitochondrial haplotypes are not shared and are diagnostically different between geographic sites (*Figure 2D*). In ADMIXTURE, the optimal number of genetic groups is six (*Figure 2C*, *Figure 2—figure supplement 2*), corresponding to the number of geographic sites, and there is basically no mixing between the six clusters (*Figure 2C*; K=6). Finally, there is isolation by distance (IBD; *Figure 2—figure supplement 3*).

Second, the sympatric ecotypes in Bergen are genetically very similar. In PCA they are not separated in the first four principal components (*Figure 2A–B*) and they are the only populations that share mitochondrial haplotypes (*Figure 2D*). In the ADMIXTURE analysis, they are only distinguished at K=7, a value larger than the optimal K. As soon as the two populations are distinguished, some individuals show signals of admixed origin (*Figure 2C*; K=7), compatible with the assumption of ongoing gene flow and incomplete reproductive isolation. These observations question the species status of *C. balticus*, which was based on the assumption of temporal isolation between these two populations (*Heimbach, 1978*).

Third, the data suggest that after the ice age *Clunio* colonized northern Europe from a single source and expanded along two fronts into the Baltic Sea and into the high Arctic. The mitochondrial haplotype network expands from a single center, which implies a quick radiation from a colonizing haplotype (*Figure 2D*; note that all sampled populations occur in areas under a former ice shield cover). There is also a large degree of shared nuclear polymorphisms. Thirty-four percent of polymorphic SNPs are polymorphic in all seven populations and 93% are polymorphic in at least two populations (*Figure 2—figure supplement 4*). Separation of the Baltic Sea populations along PC1 and the Arctic population along PC2 (*Figure 2A*), suggests that *Clunio* expanded into the high Arctic and into the Baltic Sea independently. Congruently, nucleotide diversity significantly decreases toward both expansion fronts (*Figure 2—figure supplement 5*, *Supplementary file 1*). Postglacial establishment from a common source indicates that the *Baltic* and *Arctic ecotypes* have evolved their new local adaptations during this expansion time, that is, in less than 10,000 years.

Fourth, ADMIXTURE analysis reveals that sympatric co-existence of the *Atlantic* and *Baltic ecotypes* in Bergen likely results from introgression of *Baltic ecotype* individuals into an existing *Atlantic ecotype* population. At K=2 and K=3 the two Baltic Sea populations Seh-2AR and Ar-2AR are separated from all other populations and the two Bergen populations Ber-2AR and Ber-1SL show a marked genetic contribution coming from these Baltic Sea populations (*Figure 2C*). The Baltic genetic component is slightly larger for the *Baltic ecotype* Ber-2AR population than for the *Atlantic ecotype* Ber-1SL population. A statistical analysis via TreeMix detects predominantly an introgression from Seh-2AR into Ber-2AR (*Figure 2E*, *Figure 2—figure supplements 6 and 7*), suggesting that haplotypes that convey the *Baltic ecotype* have introgressed into a population of *Atlantic ecotype*, resulting in the now observed

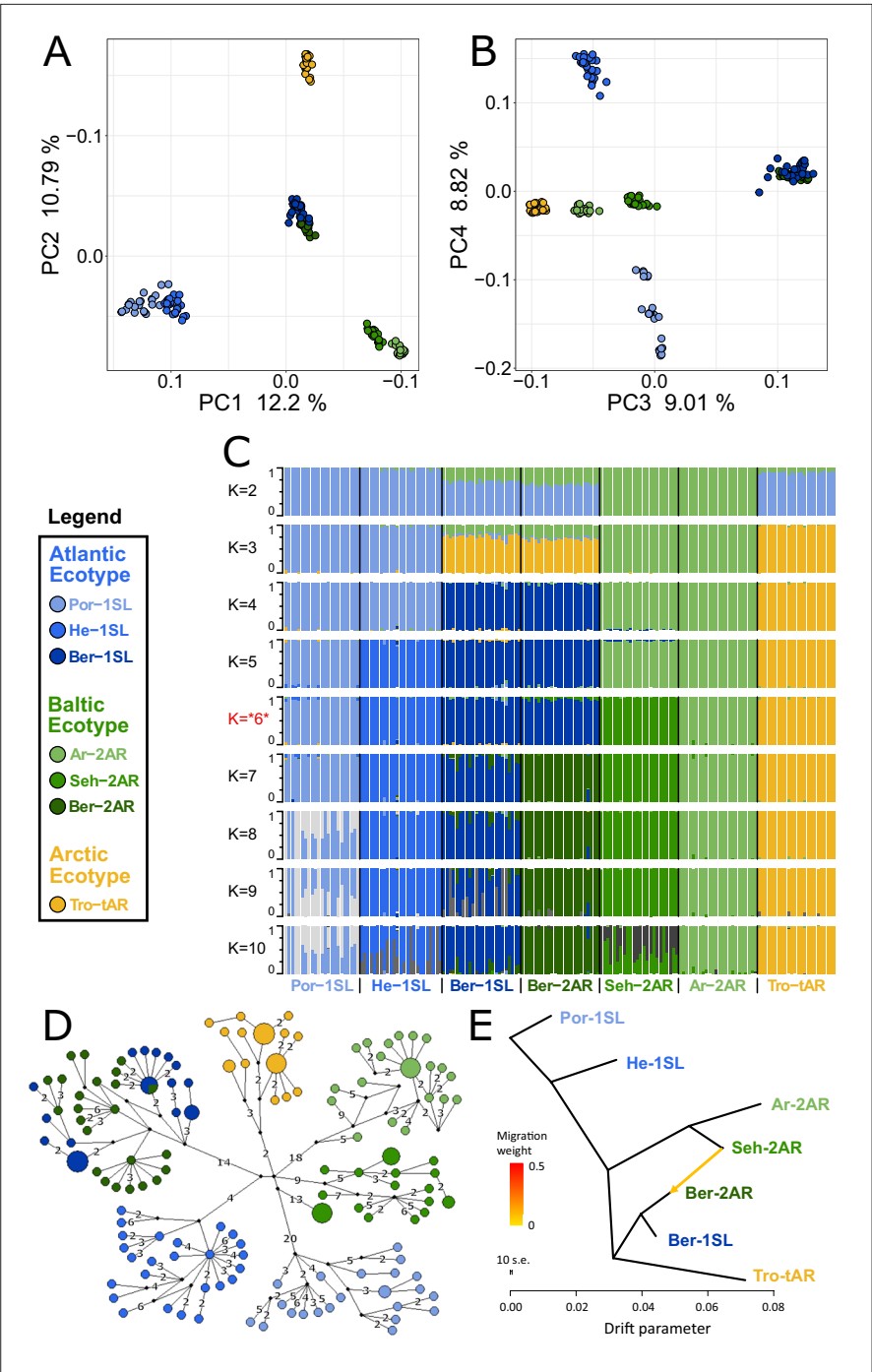

**Figure 2.** Genetic structure and evolutionary history of Northern European *Clunio* ecotypes. The analysis is based on 24 individuals from each population (23 for Por-1SL, 25 for He-1SL). (**A, B**) Principal Component Analysis (PCA) based on 792,032 SNPs separates populations by geographic location rather than ecotype. (**C**) ADMIXTURE analysis supports strong differentiation by geographic site (best K=6), but a notable genetic component from the Baltic Sea in the Bergen populations (see K=2 and 3). The Bergen populations are only separated at K=7 and then show a number of admixed individuals. (**D**) Haplotype network of full mitochondrial genomes reveals highly divergent clusters according to geographic site, but haplotype sharing between Ber-1SL and Ber-2AR. (**E**) Correlated allele frequencies indicate introgression from Seh-2AR into Ber-2AR.

The online version of this article includes the following source data and figure supplement(s) for figure 2:

**Figure supplement 1.** Principal component analysis (PCA) of all individuals for all seven populations.

*Figure 2 continued on next page*

*Figure 2 continued*

**Figure supplement 2.** The results of the cross-validation test of the ADMIXTURE analysis.

**Figure supplement 3.** Mantel test for isolation by distance between the studied populations.

**Figure supplement 3—source data 1.** Table of the input data for the Mantel test.

**Figure supplement 4.** Detected single nucleotide polymorphisms (SNPs) are largely shared between the seven populations.

**Figure supplement 5.** Nucleotide diversity π per population in 200 kb non-overlapping windows across the genome.

**Figure supplement 6.** TreeMix analysis for introgression events.

**Figure supplement 7.** Model likelihood in TreeMix analysis, dependent on the number of assumed migration events.

---

sympatric co-existence of *Baltic* and *Atlantic ecotypes* in Bergen. So far Bergen is the only site for which sympatric co-existence of the ecotypes is known. Further work will be required to see if there is a larger area of overlap, for example in the Skagerrak, and if we see similar signals of introgression or possibly secondary contact in other sympatric populations.

While our observation adds to the evidence for a role of introgression in adaptation and early speciation processes (*Edelman and Mallet, 2021*; *Baack and Rieseberg, 2007*), it also constitutes an excellent starting point for mapping loci that could be involved in the respective ecotype adaptations. Under the assumption that fast adaptation and ecotype formation are mostly based on frequency changes in standing variation, one expects that the respective haplotypes should be shared between ecotypes, especially when transferred via introgression. The following analysis is focussed on the *Atlantic* and *Baltic ecotypes*, represented by three populations each.

## Nuclear haplotype sharing

First, we reconstructed the genealogical relationship between 36 individuals (six from each population) in 50 kb windows (n=1,607) along the genome, followed by topology weighting. There are 105 possible unrooted tree topologies for six populations, and 46,656 possibilities to pick one individual from each population out of the set of 36. For each window along the genome, we assessed the relative support of each of the 105 population tree topologies by all 46,656 combinations of six individuals. We found that tree topologies change rapidly along the chromosomes (*Figure 3A*; *Figure 3—figure supplement 1*; *Supplementary file 2*). The tree topology obtained for the entire genome (*Figure 3—figure supplement 2*) dominates only in few genomic windows (*Figure 3A*, black bars 'Orig.'), while usually one or several other topologies account for more than 75% of the tree topologies (*Figure 3A*, gray bars 'Misc.'). Hardly ever do all combinations of six individuals follow a single population tree topology (*Figure 3A*, stars), which implies that in most genomic windows some individuals do not group with their population. Taken together, this indicates a massive sharing of haplotypes across populations.

In order to test whether this is only incomplete lineage sorting or if there is additional introgression beyond the detected case in the Bergen populations, we calculated the f4 ratio test for the geographically distant Por-1SL and He-1SL, as well as Ar-2AR and Seh-2AR populations. The genome-wide f4 ratio was –0.00062 (standard error 0.00008) and by comparison to coalescent simulations was found to be significantly different from 0, indicating some degree of introgression. However, the signature of introgression was driven by only 775 SNPs, which is only 0.17% of the total dataset. Thus, the largest part of haplotype sharing can be explained by incomplete lineage sorting, typical for such radiation situations.

Finally, we highlighted genomic windows that are consistent with the one clearly detected introgression event from the *Baltic ecotype* into both Bergen populations (*Figure 3A*, yellow bars 'Intr.'; all topologies grouping Por-1SL and He-1SL vs Ber-1SL, Ber-2AR, Seh-2AR, and Ar-2AR; n=357). Regions consistent with this introgression event are scattered over the entire genome.

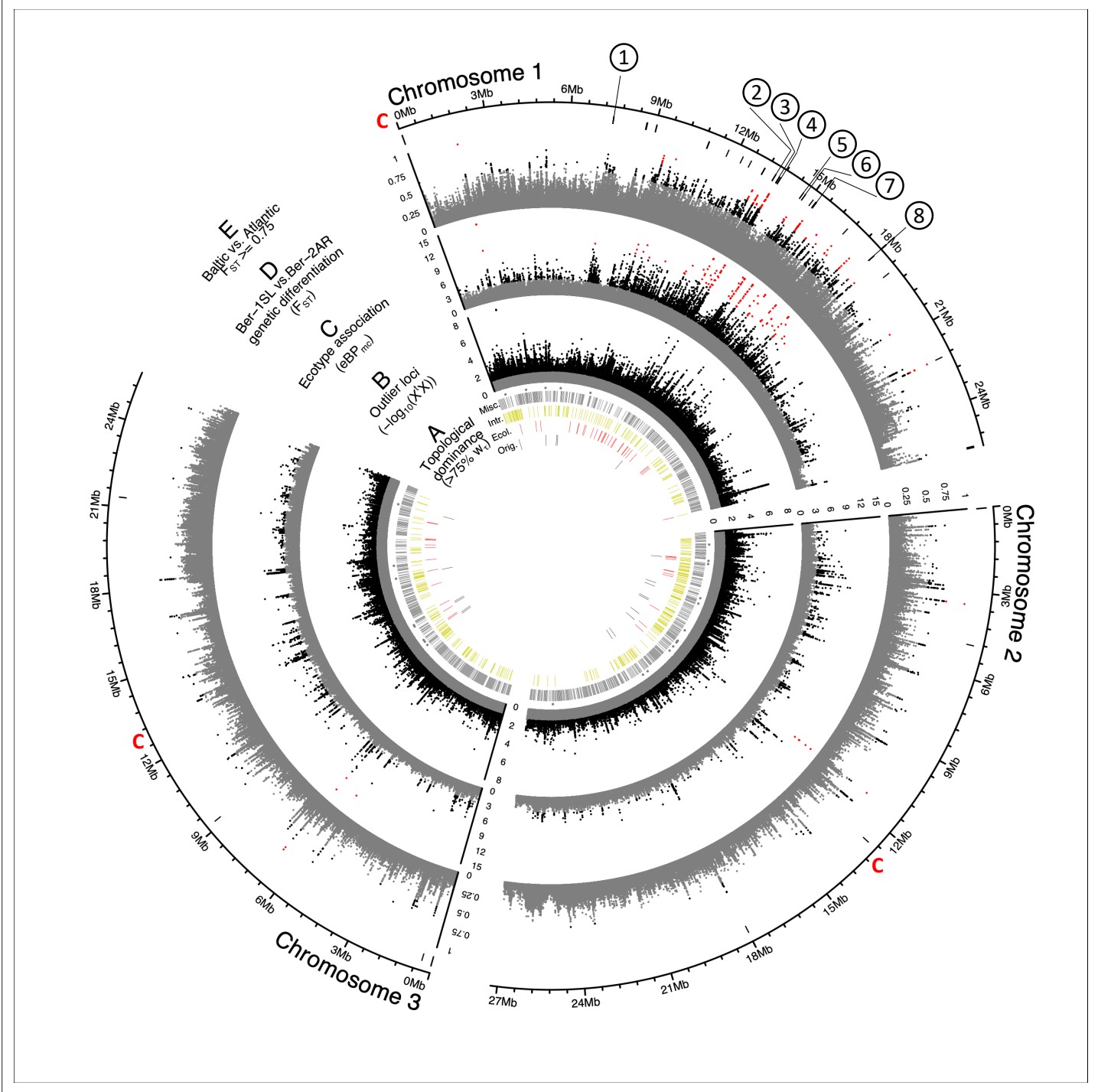

**Figure 3.** Genome screens for haplotype sharing and genotype-ecotype associations. (**A**) Topology weighting of phylogenetic trees for 36 individuals from the *Baltic* and *Atlantic ecotypes,* as obtained from 50 kb non-overlapping genomic windows. Windows were marked by a bar if they were dominated by one kind of topology ($w_t$ >75%). Most windows are not dominated by the consensus population topology ('Orig.'; *Figure 3—figure supplement 2*), but by combinations of other topologies ('Misc.'). Windows dominated by topologies that separate the *Baltic* and *Atlantic ecotypes* ('Ecol.') are mostly on chromosome 1. Windows consistent with introgression are found all over the genome ('Intr.'). (**B**) Distribution of outlier variants (SNPs and Indels) between the six *Baltic* and *Atlantic ecotype* populations, after global correction for population structure ($X^tX$ statistic). Values below the significance threshold (as obtained by subsampling) are plotted in gray. (**C**) Association of variant frequencies with *Baltic* vs. *Atlantic ecotype* ($eBP_{mc}$). Values below the threshold of 3 (corresponding to $p=10^{-3}$) are given in gray, values above 10 are given in red. (**D**) Genetic differentiation ($F_{ST}$) between the sympatric ecotypes in Bergen. Values above 0.5 are given in black, values above 0.75 in red. (**E**) The distribution of SNPs with $F_{ST} \geq 0.75$ in the *Baltic*

*Figure 3 continued on next page*

*Figure 3 continued*

vs. *Atlantic ecotypes.* Circled numbers mark the location of the eight most differentiated loci (see *Figure 6*). Centromeres of the chromosomes are marked by a red 'C'.

The online version of this article includes the following figure supplement(s) for figure 3:

**Figure supplement 1.** Incomplete lineage sorting, as illustrated by the relative support for 105 population topologies in 50 kb windows across superscaffolds 47 C and 18.

**Figure supplement 2.** Genome-wide consensus genealogy for six individuals from each of the seven populations.

**Figure supplement 3.** An overview of outlier and association analysis for ecotype (**C**), sea surface salinity (**D**) and water temperature (**E**).

**Figure supplement 4.** Pairwise $F_{ST}$ comparisons between all seven populations.

**Figure supplement 5.** Distribution of $F_{ST}$ values in all pairwise population comparisons.

**Figure supplement 6.** Genetic differentiation ($F_{ST}$) between Atlantic and Baltic ecotype.

**Figure supplement 7.** Genetic divergence ($d_{xy}$), nucleotide diversity ($\pi$) and short-range linkage disequilibrium ($r^2$) for the Ber-2AR vs. Ber-1SL comparison.

**Figure supplement 8.** Pairwise linkage disequilibrium (LD) across chromosome 1 of all Atlantic and Baltic ecotype populations.

**Figure supplement 9.** Principal component analysis (PCA) of the Atlantic and Baltic ecotypes for the highly ecotype-associated region on chromosome 1.

**Figure supplement 10.** Selection of ecotype associated genetic variants based on $X^tX$ and $eBP_{mc}$ values.

**Figure supplement 11.** The effects of ecotype-associated genetic variants on genes, as compared to the genome-wide set of genetic variants.

**Figure supplement 12.** Linkage disequilibrium (LD) decay in the studied *C.marinus* populations.

## Crosses and QTL mapping suggest that lunar rhythmicity is affected by more than one locus (Fig. 4)

As a first attempt to get hold of ecologically relevant loci, we assessed the genetic basis to lunar rhythmicity by crossing the lunar-arrhythmic Ber-2AR strain with the lunar-rhythmic Ber-1SL strain. Interestingly, the degree of lunar rhythmicity segregates within and between crossing families (*Figure 4—figure supplement 1*), suggesting a heterogeneous multi-genic basis of lunar arrhythmicity. Genetic segregation in the cross implies that the weak rhythm observed in Ber-2AR is due to genetic polymorphism. The Ber-2AR strain seems to carry some lunar-rhythmic alleles, likely due to gene flow from the sympatric *Atlantic ecotype* (see *Figure 1C* and results below).

For QTL mapping, we then picked a set of F2 families, which all go back to a single parental pair (BsxBa-F2-34; n=272; *Figure 4—figure supplement 1*) and designed a set of 23 SNP markers for QTL mapping (*Figure 4—figure supplement 2*). As a phenotype, we scored the degree of lunar rhythmicity for each individual as the number of days between the individual's emergence and day 9 of the artificial turbulence cycle, which is the peak emergence day of the lunar-rhythmic strain (*Figure 4—figure supplement 3*). However, QTL analysis with various algorithms did not reveal any significant QTL for lunar rhythmicity (*Figure 4*). This absence of evidence may partially be due to the limited number of markers and individuals, as well as a specific shortcoming in the phenotyping. Arrhythmic individuals, which by chance emerge during the peak, are recorded as rhythmic. This leads to an inevitable phenotyping error for a fraction of individuals. However, in another study, we could show that this phenotyping error does not impede the detection of an oligogenic genetic architecture in QTL mapping for a comparable arrhythmic phenotype (*Briševac et al., 2022*). As a control that in principle we had sufficient power for QTL mapping in this crossing family, we also mapped the sex determining locus, which was clearly detectable (*Figure 4*). We can conclude that lunar rhythmicity is not controlled by a single large-effect locus, but rather has an oligogenic or possibly even polygenic basis.

## Genomic regions associated with ecotype formation

Given that QTL mapping for lunar rhythmicity suggested a multi-genic basis to ecotype formation, we next applied three approaches to identify genomic regions associated with divergence between *Atlantic* and *Baltic ecotypes.*

First, genomic windows which are dominated by tree topologies that group populations according to ecotype were highlighted (*Figure 3A*, red bars 'Ecol.').

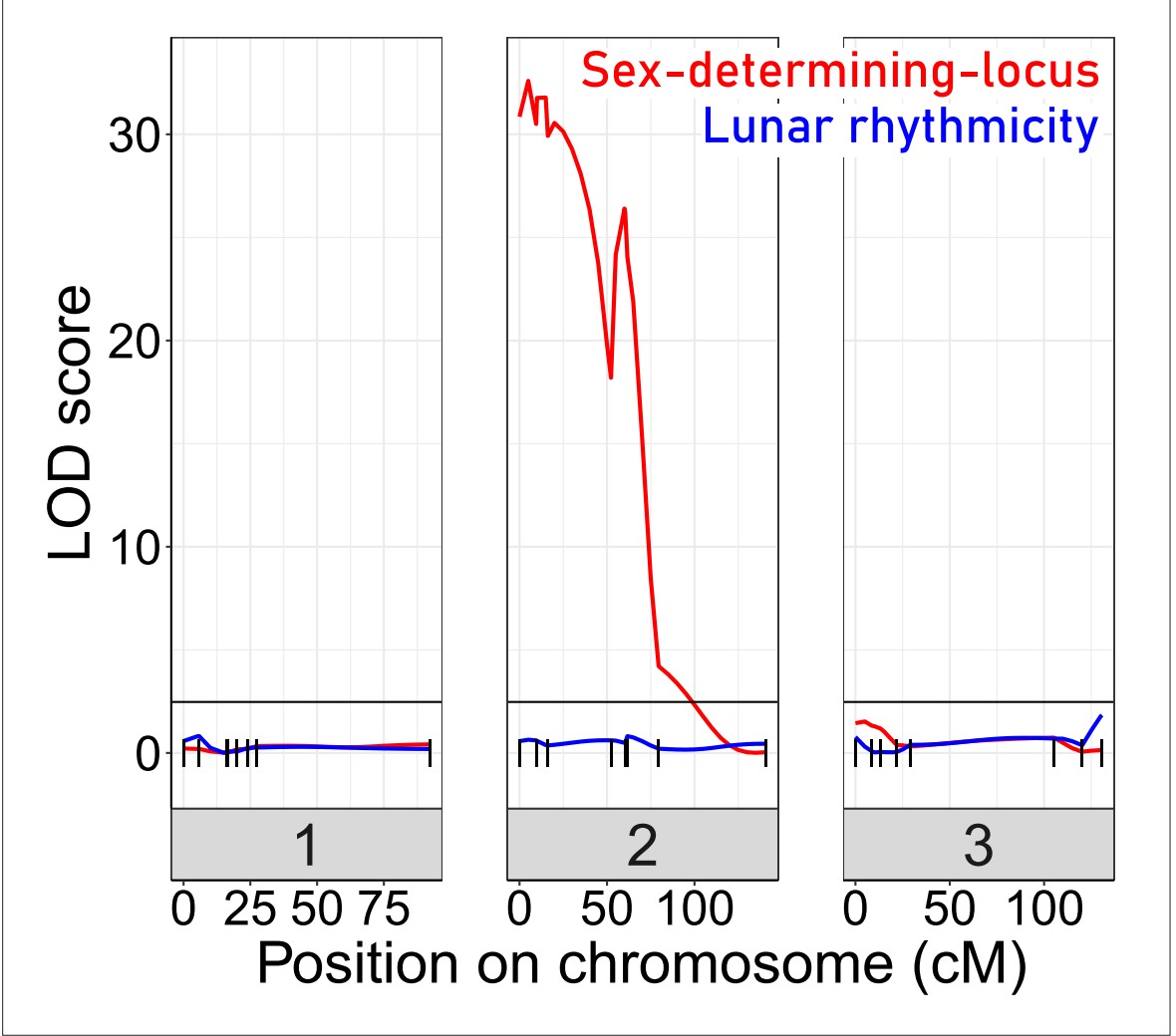

**Figure 4.** Quantitative Trait Locus (QTL) mapping for lunar rhythmicity (blue) and the sex determining locus (red). The sex-determining locus is clearly detectable (red). No significant QTLs can be detected for lunar rhythmicity, suggesting it is controlled by more than one locus. Vertical black lines are genetic markers.

The online version of this article includes the following figure supplement(s) for figure 4:

**Figure supplement 1.** Lunar emergence patterns of crosses between the Ber-1SL and Ber-2AR strains.

**Figure supplement 2.** Genetic and physical location of the designed genetic markers across the genome of *Clunio*.

**Figure supplement 3.** Phenotyping of the F2 individuals based on their emergence day.

Second, we screened all genetic variants (SNPs and indels; n=948,128) for those that are overly differentiated between the six populations after correcting for the neutral covariance structure across population allele frequencies (see $\Omega$ matrix, *Figure 3—figure supplement 3A–B*). Such variants can indicate local adaptation. At the same time, we tested for association of these variants with ecotype, as implemented in BayPass *Gautier, 2015*. Overly differentiated variants ($X^tX$ statistic; *Figure 3B*) and ecotype-associated variants ecotype (eBP$_{mc}$; *Figure 3C*) were detected all over the genome, but many were concentrated in the middle of the telocentric chromosome 1. Tests for association of variants with environmental variables such as sea surface temperature or salinity find fewer associated SNPs and no concentration on chromosome 1 (*Figure 3—figure supplement 3D–E*), confirming that the detected signals are not due to general genome properties, but are specific to the ecotypes.

Third, we expected that gene flow between the sympatric Ber-1SL and Ber-2AR populations would largely homogenize their genomes except for regions involved in ecological adaptation, which would be highlighted as peaks of genetic differentiation. While genetic differentiation may not only be due

to adaptation, but also due to demographic processes, the ecologically relevant loci should be among the differentiated loci, especially if these loci coincide with peaks in ecotype-association ($eBP_{mc}$). The distributions of $F_{ST}$ values in all pairwise population comparisons confirmed that genetic differentiation was particularly low in the Ber-1SL vs. Ber-2AR comparison (*Figure 3—figure supplements 4 and 5*). Pairwise differentiation between Ber-1SL and Ber-2AR (*Figure 3D*) shows marked peaks on chromosome 1, most of which coincide with peaks in $X^tX$ and $eBP_{mc}$. Notably, when assessing genetic differentiation of *Baltic* vs *Atlantic ecotype* (72 vs 72 individuals; *Figure 3E*; *Figure 3—figure supplement 6*), there is not a single diagnostic variant ($F_{ST} = 1$), and even variants with $F_{ST} \geq 0.75$ are very rare (n=63; *Figure 3E*).

Genetic divergence ($d_{xy}$), nucleotide diversity ($\pi$) and local linkage disequilibrium ($r^2$) of the two Bergen populations do not show marked differences along or between chromosomes (*Figure 3—figure supplement 7*). The cluster of ecotype-associated variants on chromosome 1 overlaps with three large blocks of long-range linkage disequilibrium (LD; *Figure 3—figure supplement 8*). This is also reflected in reduced linkage map length in our crossing mapping family (*Figure 4—figure supplement 2*). Reduced recombination may in part explain the clustering of divergent and ecotype-associated alleles in the middle of chromosome 1. However, the boundaries of the LD blocks do not correspond to the ecotype-associated region and differ between populations. LD blocks are not ecotype-specific. Local PCA of the strongly ecotype associated region does not reveal patterns consistent with a chromosomal inversion or another segregating structural variant (*Figure 3—figure supplement 9*). Thus, there is no obvious link between the clustering of ecotype-associated loci and structural variation. Notably, genetic differentiation is not generally elevated in the ecotype-associated cluster on chromosome 1, as would be expected for a segregating structural variant, but drops to baseline levels in between ecotype-associated loci (*Figure 3D*).

Taken together, these observations imply that numerous genomic loci – inside and outside the cluster on chromosome 1 – are associated with ecological adaptation and none of these are differentially fixed between ecotypes. In line with QTL mapping, this virtually excludes the possibility that only one or few loci have driven the new ecotype formation, suggesting instead that ecotype formation is based on a complex polygenic architecture.

## Support for adaptation from standing genetic variation

If, as assumed above, adaptation in these populations was indeed based mostly on standing genetic variation, one should find evidence of this in the data structure. We selected highly ecotype-associated SNPs ($X^tX$ >1.152094, threshold obtained from randomized subsampling; $eBP_{mc}$ >3; n=3,976; *Figure 3—figure supplement 10A*) and assessed to which degree these alleles are shared between the studied populations and other populations across Europe. Allele sharing between the Bergen populations is likely due to ongoing gene flow, and hence Bergen populations were excluded from the analysis. In turn, allele sharing between the geographically isolated Seh-2AR, Ar-2AR, Por-1SL, and He-1SL populations likely represents shared ancient polymorphism. Based on this comparison, we found that 82% of the ecotype-associated SNPs are polymorphic in both *Atlantic* and *Baltic ecotypes*, suggesting that the largest part of ecotype-associated alleles originates from standing genetic variation. We then retrieved the same genomic positions from published population resequencing data for *Atlantic ecotype* populations from Vigo (Spain) and St. Jean-de-Luz (Jean, southern France) *Kaiser et al., 2016*, an area that is potentially the source of postglacial colonization of all locations in this study. We found that 90% of the alleles associated with the Northern European ecotypes are also segregating in at least one of these southern populations, supporting the notion that adaptation in the North involves a re-assortment of existing standing genetic variation.

## Ecotypes differ mainly in the circadian clock and nervous system development

In the light of a polygenic architecture for ecotype formation, we can expect that usually several genes of relevant physiological pathways are affected. This provides an ideal setting to identify the physiological processes underlying ecotype formation by enrichment analysis. Accordingly, we assessed how all ecotype-associated variants (SNPs and indels; $X^tX$ >1.148764; $eBP_{mc}$ >3, n=4,741; *Figure 3—figure supplement 10B*) relate to *C. marinus'* genes. In a first step, we filtered the existing gene models in the CLUMA1.0 reference genome to those that are supported by transcript or protein evidence, have

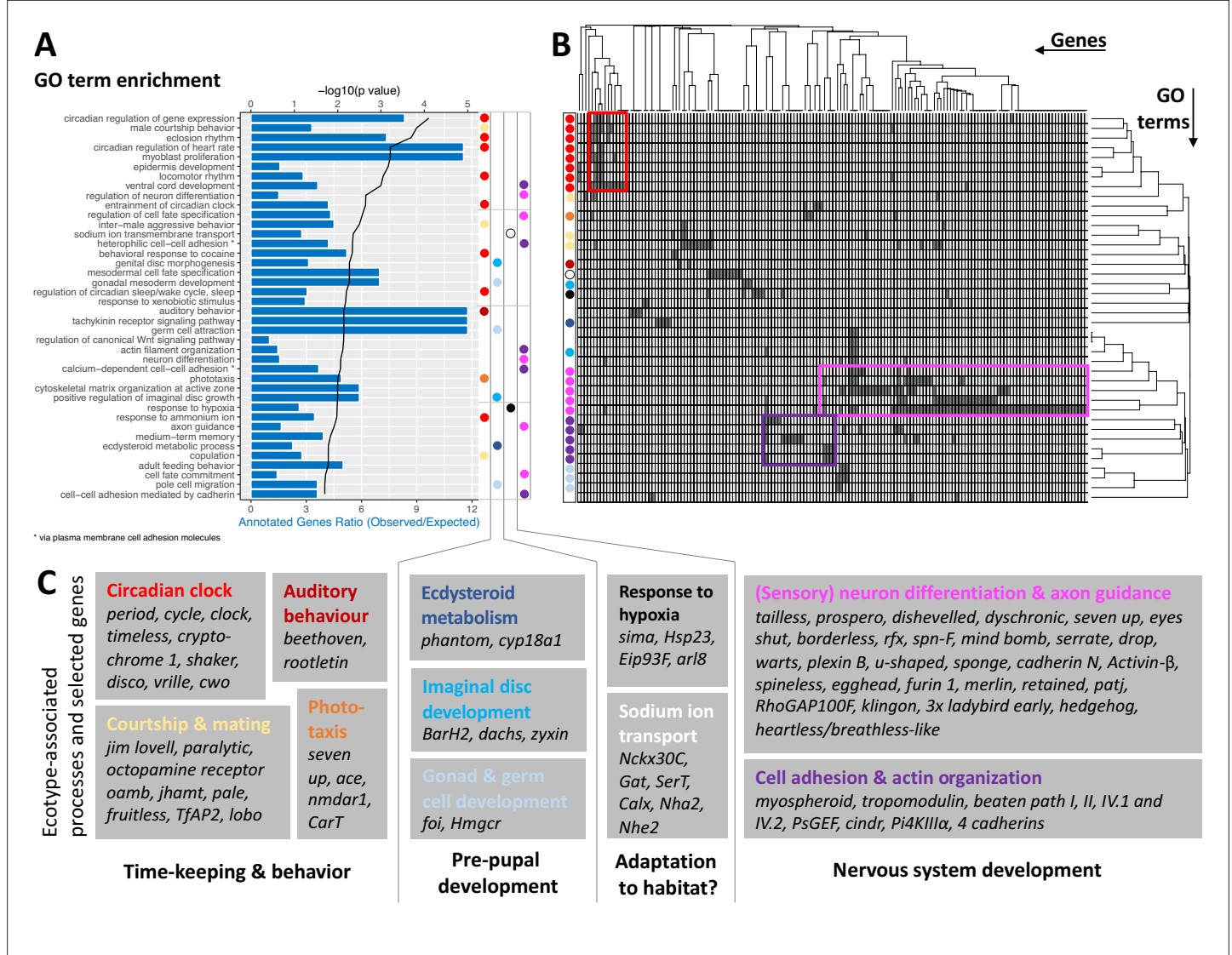

**Figure 5.** GO term analysis of ecotype associated SNPs (**A**) The top 40 enriched GO terms are listed for the 1,400 genes that are found to be affected by ecotype-associated genetic variants (eBP$_{mc}$ >3). For each GO term, the significance level (black line, top y-axis) and the observed-expected ratio of genes annotated to the respective GO term (blue bars, bottom y-axis) are given. (**B**) The top 40 GO terms are driven by 168 genes. Hierarchical clustering of genes and GO terms reveals major signals in the circadian clock and nervous system development (more details in *Supplementary file 8*). (**C**) Most GO terms are consistent with the known ecotype differences and selected genes are highlighted for all of them. Notably, basically all core circadian clock genes are affected.

known homologues or PFAM domains, or were manually curated (filtered annotations provided in *Supplementary file 3*; 15, 193 gene models). Based on this confidence gene set, we then assessed the location of variants relative to genes, as well as the resulting mutational effects (SNPeff; *Cingolani et al., 2012*; *Figure 3—figure supplement 11*; statistics in *Supplementary file 4*). The vast majority of ecotype-specific variants are classified as intergenic modifier variants, suggesting that ecotype formation might primarily rely on regulatory mutations.

The ecotype-specific SNPs are found in and around 1,400 genes (*Supplementary files 5 and 6*). We transferred GO terms from other species to the *Clunio* confidence annotations based on gene orthology (5,393 genes; see Methods and *Supplementary file 7*). GO term enrichment analysis suggests that ecological adaptation prominently involves the circadian clock, supported by three of the top four GO terms (*Figure 5A*). In order to identify which genes drive GO term enrichment in the top 40 GO terms, we extracted the genes that harbour ecotype-associated SNPs (168 genes; *Figure 5B*; *Supplementary file 8*). We individually confirmed their gene annotations and associated

GO terms. Clustering the resulting table by genes and GO terms reveals two dominant signatures (*Figure 5B*). Many GO terms are associated with circadian timing and are driven by a small number of genes, which include almost all core circadian clock genes (*Figure 5B and C*). As a second strong signal, almost half of the genes are annotated with biological processes involved in nervous system development (*Figure 5B and C*). GO term enrichment is also found for ecdysteroid metabolism, imaginal disc development, and gonad development (*Figure 5*). These processes of pre-pupal development are expected to be under circalunar clock control. The fact that circalunar clocks are responsive to moonlight and water turbulence (*Neumann, 2014*) renders the finding of GO term enrichment for 'auditory behavior' and 'phototaxis' interesting. Furthermore, many of the genes involved in nervous system development and sodium ion transport, also have GO terms that implicate them in light- and mechanoreceptor development, wiring or sensitivity (*Supplementary file 7*). With the exception of 'response to hypoxia' and possibly 'sodium ion transmembrane transport', there are very few GO terms that can be linked to the submerged larval habitat of the *Baltic ecotype*, which is usually low in salinity and can turn hypoxic in summer. There is a striking absence of GO terms involved in metabolic processes or immune system processes, despite these GO terms being annotated to 2634 genes or 185 genes, respectively.

Taken together, the detected GO terms are highly consistent with the known ecotype differences and suggest that ecotypes are mainly defined by changes in the circadian clock and nervous system development. A previously unknown aspect of *Clunio* ecotype formation is highlighted by the GO terms 'male courtship behavior', 'inter-male aggression' and 'copulation' (*Figure 5*). These processes are subject to sexual selection and considered to evolve rapidly. They could in the long term entail assortative mating between ecotypes.

## Strongly differentiated loci correspond to GO-term enriched biological processes

While GO term analysis gives a broad picture of which processes have many genes affected by ecotype-associated SNPs, this does not necessarily imply that these genes and processes also show the strongest association with ecotype. Additionally, major genes might be missed because they were not assigned GO terms. As a second line of evidence, we therefore selected variants with the highest ecotype-association by increasing the eBP$_{mc}$ cut-off to 10. This reduced the set of affected genes from 1,400 to 69 (*Supplementary files 9 and 10*). Additionally, we only considered genes with variants that are strongly differentiated between the ecotypes ($F_{ST} \geq 0.75$, compare *Figure 3E*), leaving 13 genes in 8 distinct genomic regions (*Figure 6A*; numbered in *Figure 3E*). Two of these regions contain two genes each with no homology outside *C. marinus* (indicated by 'NA', *Figure 6A*), confirming that GO term analysis missed major loci because of a lack of annotation. Three other regions contain the – likely non-visual – photoreceptor *ciliary Opsin 1* (*cOps1*) (*Velarde et al., 2005*), the transcription factor *longitudinals lacking* (*lola*; in fruit fly involved in axon guidance *Crowner et al., 2002* and photoreceptor determination *Zheng and Carthew, 2008*) and the nuclear receptor *tailless* (*tll*; in fruit fly involved in development of brain and eye *Suzuki and Saigo, 2000*), underscoring that ecotype characteristics might involve differential light sensitivity. Interestingly, *tll* also affects development of the neuroendocrine centres involved in ecdysteroid production and adult emergence *de Velasco et al., 2007*. Even more, re-annotation of this genomic locus revealed that the neighbouring gene, which is also affected by ecotype-specific variants, is the *short neuropeptide F receptor* (*sNPF-R*) gene. Among other functions, sNPF-R is involved in coupling adult emergence to the circadian clock *Selcho et al., 2017*. Similarly, only 100 kb from *cOps1* there is the differentiated locus of *matrix metalloprotease 1* (*Mmp1*), which is known to regulate circadian clock outputs via processing of the neuropeptide *pigment dispersing factor* (PDF) *Depetris-Chauvin et al., 2014*. In both cases, the close genetic linkage could possibly form pre-adapted haplotypes and entail a concerted alteration of sensory and circadian functions in the formation of ecotypes. In the remaining two loci, *sox100B* is known to affect male gonad development *Nanda et al., 2009* and the *ecdysone-induced protein 93* F is involved in response to hypoxia in flies *Lee et al., 2008*, but was recently found to also affect reproductive cycles in mosquitoes *Wang et al., 2021*. In summary, only two out of the top 13 ecotype-associated genes were included in the top 40 GO terms (*Figure 6A*). Nevertheless, all major biological processes detected in GO term analysis (*Figure 5*) are also reflected in the strongly ecotype-associated loci (*Figure 6*), giving a robust signal that circadian timing, sensory perception and nervous system development are underlying ecotype formation in *C. marinus*.

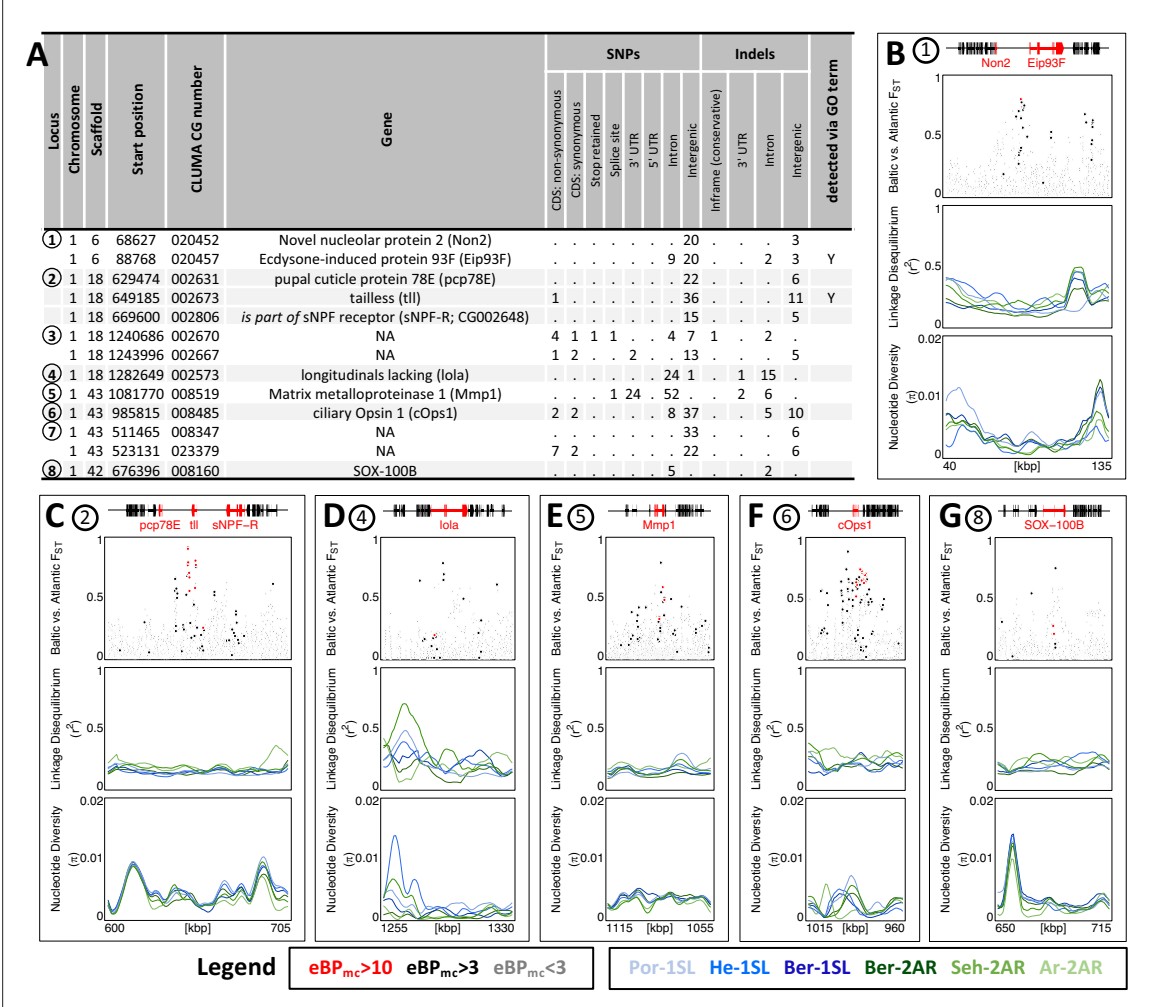

**Figure 6.** The 13 most differentiated ecotype-associated genes (**A**) Loci with highly ecotype-associated variants were selected based on eBPmc >10 and F_{ST}(Baltic-Atlantic)>0.75. There are 13 genes in eight distinct genomic loci. (**B–G**) An overview is given for the six loci with identified genes. In each panel, from top to bottom the sub-panels show the gene models, $F_{ST}$ values of genetic variants in the region, local linkage disequilibrium (LD) and genetic diversity ($\pi$). $F_{ST}$ values are colored by ecotype association of the variant (red: $eBP_{mc}$ >10; black: $10>eBP_{mc}$ > 3; gray: $eBP_{mc}$ <3). LD and genetic diversity are shown for the six populations independently, colored-coded as in *Figures 1 and 2* (10 kb overlapping sliding-windows with 1 kb steps). The are no strong signatures of selection.

The online version of this article includes the following source data for figure 6:

**Source data 1.** Table of loci with highly ecotype-associated variants.

Finally, we assessed the top 13 strongly ecotype-associated loci for signatures of selective sweeps in genetic diversity and LD (*Figure 6B–G*). Despite these loci being the most differentiated between ecotypes in the entire genome, there is at best a mild reduction in genetic diversity and a mild increase in LD (*Figure 6B–G*). If selection acted on these loci, it must have been very soft, underscoring a history of polygenic adaptation from standing genetic variation and continued recombination.

## Discussion

The deep genomic analysis of *Clunio* populations has allowed us to reconstruct a detailed biogeographic history of their different ecotypes. Most surprisingly, the two sympatric ecotypes that were considered different species turned out to be the most closely related populations in the sample, testifying the high power of ecology in generating differential adaptation in the face of gene flow. The data allowed further to infer a polygenic architecture for ecotype formation. We found that while ecotype-associated alleles differ in allele frequency, they are largely shared between ecotypes, which

suggests that adaptation primarily involves standing genetic variation from many different loci. A similar re-use of existing regulatory variation has been found in ecotype formation in sticklebacks *Chan et al., 2010*; *Kingman, 2020*; *Verta and Jones, 2019* or mimicry in *Heliconius* butterflies *Edelman et al., 2019*. However, while in *Heliconius* alleles are shared over large evolutionary distances via introgression, *Clunio* ecotypes diverged recently from a common source, as is illustrated by massive and genome-wide shared polymorphism. Combined with the observation that many genes from the same biological processes have ecotype-associated alleles, this draws a picture of polygenic adaptation, involving many pre-existing alleles with probably small phenotypic effects. Particularly for adaptation in circadian timing this scenario is highly plausible. The ancestral *Atlantic ecotype* comprises many genetically determined circadian timing types that are adapted to the local tides (*Kaiser, 2014*; *Neumann, 1967*; *Kaiser et al., 2016*; *Kaiser et al., 2021*). Existing genetic variants conveying emergence at dusk were likely selected or re-assorted to form the *Baltic ecotype*'s highly concentrated emergence at dusk.

Besides circadian timing, the ecotypes differ in circalunar timing and oviposition behavior. In our study, the vast majority of GO terms and candidate genes is consistent with these functions, leaving little risk for evolutionary 'story-telling' based on individual genes or GO terms (*Pavlidis et al., 2012*). We propose that good congruence between known phenotypic differences and detected biological processes could be a hallmark of polygenic adaptation, as only polygenic adaptation is expected to leave a footprint in many genes of the same ecologically relevant biological process.

Based on our genomic comparison of lunar-rhythmic and lunar-arrhythmic ecotypes, we propose three not mutually exclusive hypotheses on molecular pathways involved in the unknown circalunar clock. Firstly, *Clunio*'s circalunar clock is known to tightly regulate ecdysteroid-dependent development and maturation just prior to pupation (*Neumann and Spindler, 1991*). Congruently, our screen identified ecotype-associated genes in the development of imaginal discs and genital discs, and in ecdysteroid metabolism. Lunar arrhythmicity may rely on an escape of these processes from circalunar clock control. Secondly, it has been hypothesized that circalunar clocks involve a circadian clock (*Bünning and Müller, 1961*) and such a mechanism has been experimentally confirmed in the midge *Pontomyia oceana* (*Soong and Chang, 2012*). Thus, the overwhelming circadian signal in our data might be responsible for both circadian timing adaptations and the loss of circalunar rhythms. Thirdly, *Clunio*'s circalunar clock is synchronized with the external lunar cycle via moonlight, as well as tidal cycles of water turbulence and temperature (*Neumann, 2014*). Our data suggests that sensory receptor development, wiring or sensitivity might differ between ecotypes. Interestingly, some *Atlantic ecotype* populations are insensitive to specific lunar time cues, either moonlight or mechanical stimulation by the tides (*Kaiser, 2014*). These pre-existing insensitivities may have been combined to form completely insensitive and hence lunar-arrhythmic ecotypes. This scenario would fit the general pattern of polygenic adaptation through a re-assortment of standing genetic variation, which emerges from our study.

In several species, genes involved in complex behavioral or ecological syndromes were found to be locked into supergenes by chromosomal inversions, for example in *Heliconius* butterfly mimicry (*Joron et al., 2011*) or reproductive morphs of the ruff (*Küpper et al., 2016*). While we observe a clustering of ecotype-associated alleles in *Clunio*, there is no obvious connection to an underlying structural variant (SV). Possibly, the SV is so complex that it did not leave an interpretable genomic signal. Alternatively, *Clunio*'s long history of genome rearrangements (*Kaiser et al., 2016*) may have resulted in a clustering of ecologically relevant loci without locking them into a single SV. Clustering could be stabilized by low recombination, consistent with the observed three LD blocks, which – while not ecotype-specific – all overlap with the differentiated region. Epistatic interactions between the clustered loci and co-adaptation of alleles might further reduce the fitness of recombinants and lead to a concerted response to selection. Such an interconnected adaptive cluster might allow for more flexible evolutionary responses than a single, completely linked supergene. Further studies will have to show whether such a genome architecture exists, whether it facilitates adaptation and whether it might itself be selected for. Notably, a similar architecture with clustered, tightly linked loci was predicted to result from adaptation with gene flow (*Yeaman and Whitlock, 2011*), a scenario which we found in the sympatric Bergen populations.

## Methods

### Nomenclature of ecotypes

We expanded the existing naming convention of *C. marinus* timing types *Kaiser et al., 2021* to also include *Baltic* and *Arctic ecotypes*. Names of populations and corresponding laboratory strains consist of an abbreviation for geographic origin followed by a code for the daily and lunar timing phenotypes. Daily phenotypes in this study are emergence during the first 12 hr after sunrise ('1') or, emergence during the second 12 hr after sunrise ('2') or emergence during every low tide ('t' for tidal rhythm). Lunar phenotypes in this study are either emergence during full moon and new moon low tides ('SL' for semi-lunar) or arrhythmic emergence ('AR'). As a consequence, the *Arctic ecotype* is 'tAR', the *Baltic ecotype* is '2AR' and the *Atlantic ecotype* populations in this study are all of timing type '1SL' (while other timing types exist within the *Atlantic ecotype Kaiser et al., 2021*).

### Fieldwork and sample collection

Field samples for genetic analysis and establishment of laboratory strains were collected in Sehlendorf (Seh, Germany), Ar (Sweden), Tromsø (Tro, Norway), and Bergen (Ber, Norway) during eight field trips in 2017 and 2018 (*Supplementary file 11*). Field caught adult males for DNA extraction were directly collected in 99.98% ethanol and stored at –20 °C. Females are immobile and basically invisible in the field, unless found in copulation. Laboratory strains were established by catching copulating pairs in the field and transferring multiple fertilized egg clutches to the laboratory (*Supplementary file 11*). Samples and laboratory strains of the sympatric ecotypes in Bergen were collected at the same location but at different daytime. Additional samples and laboratory strains from Helgoland (He, Germany) and Port-en-Bessin (Por, France) were collected and described earlier *Kaiser et al., 2016*; *Kaiser et al., 2021*; *Kaiser et al., 2010*, but had previously not been subject to whole genome sequencing of individuals.

### Laboratory culture and phenotyping of ecotypes

Laboratory strains were reared under standard conditions *Neumann, 1966* at 20 °C with 16 hr of light and 8 hr of darkness. *Atlantic* and *Arctic ecotype* strains were kept in natural seawater diluted 1:1 with deionized water and fed with diatoms (*Phaeodactylum tricornutum*) and powdered nettles (*Urtica sp.*). The *Baltic ecotype* was kept in natural Sea water diluted 1:2 and fed with diatoms and powdered red algae (90%, *Delesseria spp.*, 10% *Ceramium spp.*, obtained from F. Weinberger and N. Stärck, GEOMAR, Kiel). For entrainment of the lunar rhythm all strains were provided with 12.4 hr tidal cycles of water turbulence (mechanically induced vibrations produced by an unbalanced motor, 50 Hz, roughly 30 dB above background noise, 6hr 10 min on, 6 hr 15 min off) (*Neumann and Heimbach, 1979*; *Neumann, 1978*).

Assignment of strains to ecotypes was confirmed based on their phenotypes as recorded in laboratory culture. Oviposition behavior was assessed during standard culture maintenance: *Baltic ecotype* eggs are generally found submerged at the bottom of the culture vessel, *Atlantic* and *Arctic ecotype* eggs are always found floating on the water surface or on the walls of the culture vessel (for details see next paragraph). Daily emergence times were recorded in 1 hr intervals by direct observation (Seh-2AR, Ar-2AR) or with the help of a fraction collector (*Honegger, 1977*) (Ber-1SL, Ber-2AR, Tro-tAR, Por-1SL, He-1SL; *Figure 1—figure supplement 1*). Lunar emergence times were recorded by counting the number of emerged midges in the laboratory cultures every day over several months and summing them up over several tidal turbulence cycles. Emergence data for He-1SL was taken from *Neumann, 1983*, emergence data for Por-1SL was taken from *Briševac et al., 2022*.

### Assessment of oviposition behavior

Because of the lack of tides in the Baltic Sea, the Baltic ecotype cannot rely on the tides to expose the larval substrates for egg deposition. Therefore, in contrast to the Atlantic and Arctic ecotypes, the Baltic ecotype does not oviposit on exposed larval substrates, but on the water surface, from where the eggs sink to the bottom of the sea. This change in oviposition preference is accompanied by a specific behavioral change, namely the bending of the female's abdomen down through the water surface *Endraß, 1976b*, which ensures that the egg masses are not caught in the water's surface tension. In our laboratory culture, the animals are kept in plastic boxes with a constant water level, so that larval substrates are never exposed. Under these conditions, most of the Baltic ecotype's

egg masses will be found submerged at the bottom of the culture box, as expected. Very rarely egg masses are deposited on the walls or the lid of the box. The Atlantic and Arctic ecotypes will – in the absence of exposed larval substrates – also oviposit on the water surface. However, as they lack the characteristic banding of the female abdomen during oviposition, their egg masses will be trapped in the water's surface tension and cannot sink to the bottom of the culture box. Their egg masses will always float on the water surface, usually with the female still sticking to the egg jelly. These floating egg masses often form aggregations on the water surface. We used these differences in where the egg clutches are found in our laboratory cultures to additionally confirm ecotype identity of our laboratory strains.

## Crosses and quantitative trait locus (QTL) mapping

The Ber-1SL and Ber-2AR laboratory strains were subject to single-pair crossing after their emergence rhythms were synchronized by keeping them in separate, time-shifted LD regimes. Several F1 families were raised independently and the F1 siblings were allowed to mate freely in order to obtain bulk F2 families. Emergence distributions were recorded by collecting all emerged adults every day.

Bulk crossing family BsxBa-F2-34 (n=272) was selected for QTL mapping. DNA of the two parents and the F2 progeny was extracted with the QuickExtract DNA Extraction Solution (Lucigen, QE0905T) according to manufacturer's protocol with modifications. Instead of vortexing before incubation, a pestle was used to grind the sample. In order to identify scorable markers for the genetically very similar strains, the two parents of the cross were subject to whole-genome sequencing. Raw reads were processed through our genotyping pipeline (described below), resulting in 656,368 detected variants, 53,097 of which were diagnostic for the grandparents. Based on the list of diagnostic markers, we picked eight evenly spaced regions (physical distance) on each chromosome for amplicon sequencing (*Figure 4—figure supplement 2*, *Supplementary file 12*). Multiplex PCR was performed with the QIAGEN Multiplex PCR Kit (206143) for each chromosome primer set separately (15 min at 95 °C; 40 cycles of 94 °C for 30 s, 57 °C for 90 s and 72 °C for 90 s; 72 °C for 10 min). Sequencing libraries were prepared with standard Illumina protocols and sequenced with 150 bp paired-end on the Illumina HiSeq3000 sequencer by the Max Planck Genome Centre (Cologne, Germany). Sequencing reads from independent runs were first merged to one read file for forward and reverse reads. Those files were again subject to our genotyping pipeline (described below) with the exception that the mapped reads in SAM format undergone a coverage capping of 40 using a custom python script and the VCF was filtered for a minimum minor allele frequency of 0.25 ('--maf 0.25') and a maximum proportion of missing data per locus of 30% ('--max-missing 0.7'). All markers that were not diagnostic for the parents were removed. In case genotypes differed within an amplicon, a consensus amplicon genotype was called by majority.

For QTL mapping, the degree of rhythmicity of an individual was recorded as the number of days between its emergence day and day 9, which is the peak of emergence in the rhythmic strain (Ber-1SL). The phenotyping has a specific, inevitable shortcoming: Arrhythmic individuals can also emerge during the peak days by chance, so that this small fraction of individuals is phenotyped as rhythmic while actually being arrhythmic. As the phenotype is not normally distributed (Kolmogorov-Smirnov test and the Shapiro-Wilk test of normality) it was treated as non-parametric. QTL scans were performed in the R package 'qtl' (*Broman et al., 2003*). The genome scans with different QTL models followed the calculation of conditional genotype probabilities in 5 cM distances and simulated genotypes from 64 imputations with the same distance. Both phenotypes (lunar rhythmicity, sex) were scanned for a single QTL model in the 'scanone()' function (method = 'em'). Significance thresholds were set at top 5% of the logarithm of the odds (LOD) scores for 1000 permutations. The 'stepwise. qtl()' function for identification of multiple and interacting QTLs returned a null QTL model for lunar rhythmicity.

## DNA extraction and whole genome sequencing

For each of the seven populations, 24 field caught males (23 for Por-1SL, 25 for He-1SL) were subject to whole genome sequencing. DNA was extracted from entire individuals with a salting out method (*Reineke et al., 1998*) and amplified using the REPLI-g Mini Kit (QIAGEN) according to the manufacturer's protocol with volume modifications (*Supplementary file 13*). All samples were subject to whole genome shotgun sequencing at 15–20 x target coverage on an Illumina HiSeq3000 sequencer

with 150 bp paired-end reads. Library preparation and sequencing were performed by the Max Planck Genome Centre (Cologne, Germany) according to standard protocols. Raw sequence reads are deposited at ENA under Accession PRJEB43766.

## Sequence data processing, genotyping, and SNP filtering

Raw sequence reads were trimmed for adapters and base quality using Trimmomatic v.0.38 *Bolger et al., 2014* with parameters 'ILLUMINACLIP:TruSeq3-PE-2.fa:2:30:10:8:true', 'LEADING:20', 'TRAILING:20', 'MINLEN:75'. Overlapping paired end reads were merged with PEAR v.0.9.10 *Zhang et al., 2014*, setting the minimum assembled sequence length to 75 bp and a capping quality score of 20. Assembled and unassembled reads were mapped with BWA-MEM *Li, 2013* to the nuclear reference genome *Kaiser et al., 2016* (ENA accession GCA_900005825.1) and the mitochondrial reference genome (ENA accession CVRI01023763.1) of *C. marinus*. Mapped reads were sorted, indexed, filtered for mapping quality (-q 20) and transformed to BAM format with SAMtools v.1.9 *Li et al., 2009*. Read group information was added with the AddOrReplaceReadGroups.jar v.1.74 script from the Picard toolkit (http://picard.sourceforge.net/) *DePristo et al., 2011*.

For the nuclear genome, SNP and insertion-deletion (indel) genotypes were called using GATK v.3.8–0-ge9d806836 *McKenna et al., 2010*. After initial genotype calling with the GATK Haplotype-Caller and the parameter '-stand_call_conf 30', base qualities were recalibrated with the GATK BaseRecalibrator with '-knownSites' and genotype calling was repeated on the recalibrated BAM files to obtain the final individual VCF files. Individual VCF files were combined using GATK GenotypeGVCFs. SNP and indel genotypes were filtered with VCFtools v.0.1.14 *Danecek et al., 2011* to keep only biallelic polymorphisms (--max-alleles 2), with a minimum minor allele frequency of 0.02 (--maf 0.02), a minimum genotype quality of 20 (--minQ 20) and a maximum proportion of missing data per locus of 40% (--max-missing 0.6), resulting in 792,032 SNPs and 156,096 indels over the entire set of 168 individuals. For certain analyses indels were excluded with VCFtools ('--remove-indels').

Reads mapped to the mitochondrial genome were transformed into mitochondrial haplotypes as described in *Fuhrmann and Kaiser, 2020*.

## Population genomic analyses

Mitochondrial haplotype networks were calculated using the Median-Joining algorithm *Bandelt et al., 1999* with Network v.10.1.0.0 (fluxus-engineering.com).

Nuclear SNP genotypes were converted to PLINK format with VCFtools. SNPs were LD pruned with PLINK v.1.90b4 *Chang et al., 2015* and parameters '--indep-pairwise 50 10 0.5' as well as '--chr-set 3 no-xy no-mt --nonfounders' (resulting in 445,385 SNPs). Principal Component Analysis (PCA) was performed in PLINK using the option '--pca' with the options default settings. The pruned BED file from PLINK was used as input to ADMIXTURE v.1.3.0 *Alexander and Lange, 2011*, with which we assessed a series of models for K=1 to K=10 genetic components, as well as the corresponding cross-validation error ('--cv'). Migration was further tested by converting the SNP data to TreeMix format with the *vcf2treemix.sh* script *Ravinet, 2021* and running *TreeMix 1.13 Pickrell and Pritchard, 2012* with default parameters and the southernmost Por-1SL as root population. Finally, in order to distinguish incomplete lineage sorting from introgression on a broader geographic scale, we calculated the f4 ratio test *Reich et al., 2009* as implemented in the f4.py script *Meyer et al., 2017* for the He-1SL, Por-1SL, Ar-2AR and Seh-2AR populations. Standard error was obtained by jack-knifing in blocks of 100 SNPs (-k 100). We tested if the observed f4 ratio was significantly different from zero by comparison to 1,000 coalescent simulations on a random subset of 1% of the pruned SNPs (n=4,453; -s 1000).

Population estimates along the chromosomes were calculated in 100 kb overlapping sliding-windows with 10 kb steps (*Figure 3—figure supplement 7*) or 10 kb overlapping sliding-windows with 1 kb steps (*Figure 6*). Nucleotide diversity ($\pi$) was calculated for SNPs with VCFtools '--window-pi'. For the genome-wide average, calculations were repeated with 200 kb non-overlapping windows. Linkage disequilibrium (LD; as $r^2$) was calculated in VCFtools with '--geno-r2'. Local LD was calculated with '--ld-window-bp 500'. Preliminary tests showed that local LD decays within a few hundred base pairs (*Figure 3—figure supplement 12*). For long range LD, minor allele frequency was filtered to 0.2 ('--maf 0.2', resulting in 335,800 SNPs), only values larger 0.5 were allowed with '--min-r2 0.5' and the '--ld-window-bp 500' filter was removed. Pairwise $F_{ST}$ was calculated with VCFtools '--weir-fst-pop' option per SNP and in sliding windows. For calculation of genetic divergence ($d_{xy}$), allele frequencies

were extracted with VCFtools '--freq' and $d_{xy}$ was estimated from allele frequencies according to *Delmore et al., 2015*.

## Phylogenomics and topology weighting

Nuclear genome phylogeny was calculated for a random set of six individuals from each population, without Tro-tAR (n=36). We down-sampled to six individuals per population because the following topology weighting algorithm assesses all possible combinations of six individuals (i.e. one per population) in the sample. The number of combinations grows exponentially with the number of individuals and thus leads to immense run-times of the algorithm. For windowed phylogenies, the VCF file was subset into non-overlapping 50 kb windows using VCFtools '--from-bp --to-bp'. SNP genotypes were transformed into FASTA alignments of only informative sites with the vcf2phylip.py v.2.3 script *Ortiz, 2019* and parameters '-m 1 p -f'. Heterozygous genotypes were represented by the respective IUPAC code for both bases. Whole genome and windowed phylogenies were calculated with IQ-TREE v.1.6.12 *Nguyen et al., 2015* using the parameters '-st DNA -m MFP -keep-ident -redo' for the windowed and '-st DNA -m MFP -keep-ident -bb 1000 -bnni -nt 10 -redo' for the whole genome phylogenies. Topology weighting was performed on the windowed phylogenies with TWISST *Martin and Van Belleghem, 2017* and the parameter '--method complete'.

## Association analysis

Population-based association between genetic variants (SNPs and Indels) and ecotype, as well as environmental variables (*Supplementary file 14*) was assessed in BayPass v.2.2 *Gautier, 2015*. Allele counts were obtained with VCFtools option '--counts'. Analyzed covariates were ecotype, sea surface salinity (obtained from *Hordoir et al., 2019*) and average water temperature of the year 2020 (obtained from weather-atlas.com, accessed 27.04.2020; 16:38), as given in *Supplementary file 14*. BayPass was run with the MCMC covariate model. BayPass corrects for population structure via $\Omega$ dissimilarity matrices, then calculates the $X^tX$ statistics and finally assesses the approximate Bayesian p value of association ($eBP_{mc}$). To obtain a significance threshold for $X^tX$ values, the data was randomly subsampled (100,000 genetic variants) and re-analyzed with the standard covariate model, as implemented in baypass_utils.R. All analyses we performed in three replicates (starting seeds 5,001, 24,306, and 1,855) and the median is shown.

## SNP effects and GO term enrichment analysis

Gene annotations to the CLUMA1.0 reference genome *Kaiser et al., 2016* were considered reliable if they fulfilled one of three criteria: (1) Identified ortholog in UniProtKB/Swiss-Prot or non-redundant protein sequences (nr) at NCBI or PFAM domain, as reported in *Kaiser et al., 2016*. (2) Overlap of either at least 20% with mapped transcript data or 40% with mapped protein data, as reported in *Kaiser et al., 2016*. (3) Manually annotated. This resulted in a 15,193 confidence genes models. The location and putative effects of the SNPs and indels relative to these confidence gene models were annotated using SnpEff 4.5 *Cingolani et al., 2012* (build 2020-04-15 22:26, non-default parameter `-ud 0'). Gene Ontology (GO) terms were annotated with emapper-2.0.1. *Huerta-Cepas et al., 2017* from the eggNOG 5.0 database *Huerta-Cepas et al., 2019*, using DIAMOND *Buchfink et al., 2015*, BLASTP e-value $<1e^{-10}$ and subject-query alignment coverage of >60%. Conservatively, we only transferred GO terms with 'non-electronic' GO evidence from best-hit orthologs restricted to an automatically adjusted per-query taxonomic scope, resulting in 5,393 *C. marinus* gene models with GO term annotations. Enrichment of 'Biological Process' GO terms in the genes associated with ecotype-specific polymorphisms was assessed with the weight01 Fisher's exact test implemented in topGO *Alexa and Rahnenfuhrer, 2010* (version 2.42.0, R version 4.0.3). When constructing the topGOdata object, the 5393 gene models with GO terms were specified as the custom background universe by providing the gene list to the parameter 'allGenes' and the custom gene-to-GO mappings were supplied by providing eggNOG GO annotations of these genes to the parameter 'gene2GO'. Out of the 1,400 ecotype-associated gene models 463 were annotated with GO terms and only those went into GO term enrichment analysis. The fractions of genes with missing GO terms among the ecotype-associated genes and the total genes do not differ significantly (chi square test, p=0.12), suggesting both gene sets are equally affected by missing GO terms. Any potential biases in the reduced gene universe are accounted for by calculating the

probability of observing at least the same degree of overrepresentation of a GO term when 463 genes are randomly selected out of the specified gene universe of 5393 genes (as implemented with the Fisher's exact test in topGO).

## Figure preparation

Figures were prepared in R *Crawley, 2007*. Data were handled with the 'data.table' *Dowle, 2020* and 'plyr' *Wickham, 2011* packages. The map of Europe was generated using the packages 'ggplot2' *Wickham, 2016* and 'ggrepel' *Slowikowski, 2018*, 'maps' *Brownrigg et al., 2018* and 'mapdata' *Becker et al., 2018*. The map was taken from the CIA World DataBank II (http://www.evl.uic.edu/pape/data/WDB/). Circular plots were prepared using the R package 'circlize' *Gu et al., 2014*. Multiple plots were combined in R using the package 'Rmisc' *Hope, 2013*. The graphical editing of the whole genome phylogeny was done in Archeopteryx (http://www.phylosoft.org/archaeopteryx) *Han and Zmasek, 2009*. Final figure combination and graphical editing of the raw plot files was done in *Inkscape*. Neighbor Joining trees of the omega statistic distances from BayPass were created with the R package 'ape' *Paradis and Schliep, 2019*. In all plots, the order and orientation of scaffolds within the chromosomes follows the published genetic linkage map *Kaiser et al., 2016*.

## Acknowledgements

For field work we obtained logistic support from the Ar Research Station (Uppsala University), the Marine Biological Station Espegrend (University of Bergen), Even Jørgensen (The Arctic University of Norway, Tromsø) and Florian Weinberger and Nadja Stärck (GEOMAR Helmholtz Centre for Ocean Research, Kiel). We thank Jürgen Reunert, Kerstin Schäfer and Susanne Mentz for technical assistance, as well as all members of the MPRG "Biological Clocks" for discussion and support. Diethard Tautz and Julien Dutheil critically read the manuscript. Whole genome sequencing was performed at the Max Planck Genome Center (Cologne) with financial support from the Max Planck Society. This work was funded by the Max Planck Society through the Max Planck Research Group "Biological Clocks" and a sequencing grant. The work was further funded by the European Research Council (ERC) under the Horizon 2020 research and innovation program with an ERC Starting Grant (Grant agreement 802923) awarded to TSK.

## Additional information

### Funding

| Funder | Grant reference number | Author |
| --- | --- | --- |
| European Research Council | 802923 | Tobias S Kaiser |
| Max-Planck-Gesellschaft | | Tobias S Kaiser |

The funders had no role in study design, data collection and interpretation, or the decision to submit the work for publication. Open access funding provided by Max Planck Society.

### Author contributions

Nico Fuhrmann, Data curation, Formal analysis, Investigation, Visualization, Writing - review and editing, Field work; Celine Prakash, Formal analysis, Investigation, Visualization, Writing - review and editing; Tobias S Kaiser, Conceptualization, Resources, Formal analysis, Supervision, Funding acquisition, Investigation, Visualization, Writing – original draft, Project administration

### Author ORCIDs

Tobias S Kaiser http://orcid.org/0000-0002-4126-0533

### Decision letter and Author response
Decision letter https://doi.org/10.7554/eLife.82824.sa1
Author response https://doi.org/10.7554/eLife.82824.sa2

## Additional files

### Supplementary files

• Supplementary file 1. Wilcoxon rank sum test with continuity correction for significant differences in nucleotide diversity (π) between populations, based on the arithmetic mean of 200 kb genomic windows. The lower-left part of the table shows the p-values and the upper-right part the corresponding significance levels: **** - 0–0.0001; *** - 0.0001–0.001; ** - 0.001–0.01; * - 0.01–0.05; ns - 0.05–1.

• Supplementary file 2. Twisst output.

• Supplementary file 3. Filtered gene annotations for the CLUMA1.0 genome assembly.

• Supplementary file 4. SnpEff analysis for selected and all variants. SnpEff was run on the selected BayPass ecotype-associated variants and on all variants. For each analytical group (Region, Impact, Function, Variant) the total numbers for each subgroup and the fractions in respect to the entire analytical group are given. Additionally, the p-values for significant deviation between the selected and entire genome variants were calculated using Fisher's exact test.

• Supplementary file 5. Genes with eBPmc larger than 3.

• Supplementary file 6. SNPeff results for genes with eBPmc larger than 3.

• Supplementary file 7. GO term annotations for the CLUMA1.0 genome assembly.

• Supplementary file 8. List of genes with ecotype-associated variants, which drive the top 40 GO terms in GOterm enrichment analysis. Gene identity was individually confirmed (see column "Gene_Name_curated"). The order of genes and GO terms corresponds to *Figure 4B*.

• Supplementary file 9. Genes with eBPmc larger than 10.

• Supplementary file 10. SNPeff results for genes with eBPmc larger than 10.

• Supplementary file 11. Sampling sites and sampling campaigns for the five newly established laboratory strains in this study. On sampling dates in squared brackets egg clutches for setting up laboratory cultures were collected from copulating pairs on the water surface. Samples from underlined dates were used for the genomic analyses of the wild populations.

• Supplementary file 12. Multiplex PCR primer pairs for QTL mapping between the sympatric populations Ber-1SL and Ber-2AR. All primers were designed and tested by Kerstin Schaefer.

• Supplementary file 13. Volume modifications for the REPLI-g Mini Kit QIAGEN 150025 whole genome amplification kit (QIAGEN). MM – Master Mix.

• Supplementary file 14. The three covariates used for association analysis in BayPass.

• MDAR checklist

### Data availability

Sequencing data for field samples and mapping families are deposited at ENA under Accession PRJEB43766. Filtered genome annotations are given in Supplementary File 3. All other data are included in the manuscript as supplementary data files.

The following dataset was generated:

| Author(s) | Year | Dataset title | Dataset URL | Database and Identifier |
|---|---|---|---|---|
| Fuhrmann N, Prakash C, Kaiser TS | 2022 | Baltic and Arctic Clunio, whole genome resequencing | https://www.ebi.ac.uk/ena/browser/view/PRJEB43766 | European Nucleotide Archive, PRJEB43766 |

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
