## [Editor Report]

This valuable study combines phenotypic analysis, quantitative genetics and population genomics to provide solid evidence for multiple genes underlying adaptive divergence in a marine midge system linked to tidal rhythm. Genes with a plausible role in perceiving and responding to lunar information are among the loci that most highly differentiate populations with distinct behaviors.

---

## [Decision Letter]

**Decision letter after peer review:**

Thank you for submitting your article "Polygenic adaptation from standing genetic variation allows rapid ecotype formation" for consideration by *eLife*. Your article has been reviewed by 2 peer reviewers, and the evaluation has been overseen by a Reviewing Editor and Detlef Weigel as the Senior Editor. The reviewers have opted to remain anonymous.

The reviewers have discussed their reviews with one another, and the Reviewing Editor has drafted this to help you prepare a revised submission. As you will see, both expert reviewers like the topic and the system you have established, but have also significant concerns, mostly about interpretation, but some also regarding technical issues. The companion paper is clearly the stronger paper, but we would like nevertheless invite revision for the current paper as well, as the two papers complement each other.

Essential revisions:

1) Describe the power of the QTL analysis and clearly describe its limitation. State that it remains unclear how complex the genetic basis of rhythmic versus arrhythmic behavior is, and that a moderately complex genetic basis (vulgo oligogenic) cannot be excluded.

2) Be more open to the possibility that some of the apparent differentiation is due to demographic processes instead of adaptation only.

3) Be more conservative with the GO enrichment analyses.

*Reviewer #1 (Recommendations for the authors):*

The Introduction should set the scene better before moving on to Clunio. Other cases of polygenic adaptation should be mentioned. There should be some discussion of theoretical expectations for the distribution of effect sizes for adaptive traits (with or without gene flow – see Yeaman and Whitlock 2011, for example).

The interpretations are generally strong and the core points are interesting and significant for the field – the recent, polygenic adaption from standing variation, the genomic clustering and the identification of candidate pathways are all strong results. Evidence from the QTL analysis is least convincing – see specific comments below. The absence of evidence here is consistent with the results of the genome scan but that is about as far as one can go.

Absence of evidence for structural variants is also quite hard to interpret. I agree that the patterns do not obviously fit with a simple interpretation (such as a single polymorphic inversion) but there is remarkably strong LD over long distances. This might be explained by gene flow for the sympatric population pair but it appears to extend to other populations too. That said, I think the authors handle this well, leaving open the possibility that further work will reveal structural variants.

Argument for standing variation does seem good – question about maintenance – plausibly suggest LA within the ancestral Atlantic ecotype.

The GO enrichment results are very neat. A key strength is knowing the dominant axis of selection and being able to link that to genetic pathways likely to influence the relevant phenotypes (and so avoid 'story-telling'). These points are made in the Discussion but I think some of this could usefully be moved to a revised introduction, perhaps giving more emphasis to insights other than the polygenic nature of adaptation that can come from the unique features of this system.

Below I give further comments by line number:

15 – Not 'is' but 'can be' since speciation by other routes is possible and also because ecotype formation need not lead to speciation.

36 – In addition to these recent references, it would be good to recognise long-standing awareness of the role of polygenic adaptation. See, for example, Le Corre and Kremer 2012 – doi: 10.1111/j.1365-294X.2012.05479.x

38 – A polygenic basis can be demonstrated by quantitative genetics methods such as 'regional heritability'. It is currently unclear whether this is included under 'deep genomic and genetic analysis'.

Figure 1 – The meaning of zero on the x-axis needs to be explained in the figure caption. The y-axis label (% emergence) seems to conflicts with the legend which says that these are plots of oviposition timings. Bergen is very far from the entrance to the Baltic – somewhere, the ecotype present in the Skagerrak needs to be mentioned.

118 – I am not sure why this is described as a contrast. The previous paragraph describes geographic rather than ecotype structure. That is surely confirmed by the observation that sympatric populations of different ecotypes are similar genetically.

154-164 – This inference should not be pushed too far since it is based on a single point in what is, presumably, a large area of overlap, perhaps with varying past and present mixes of the two ecotypes. So, the inferences apply to the specific sampled populations but may not generalise to the Atlantic-Baltic interaction at other locations. It also begs the question of whether this is a secondary contact, after an allopatric origin of the Baltic ecotype.

190 – How many topologies fit this pattern?

210 – Comparison to the SD locus only says that the number of loci for rhythmicity is likely to be >1, and even this conclusion is questionable since sex is a binary phenotype with no environmental variance. Therefore, to conclude from the lack of detectable QTL that 'many' loci underlie rhythmicity is going too far. It may be possible to use simulation to get a better idea of the power of the QTL analysis and so the likely maximum effect size. If the vertical lines in Figure 4 represent marker loci (not specified in the legend) then the number is rather low and distribution rather uneven, perhaps making QTL detection in some parts of the genome difficult (including the important region in the centre of Chr1). I also wonder how good a measure of rhythmicity is provided by the emergence day (as an absolute deviate, if I understand correctly) for an individual. This seems like a difficult trait to handle since rhythmicity cannot easily be measured for any one individual.

280 – It is surprising that the cluster of outliers on Chr1 does not appear as a QTL, especially given the long-range LD in this area

Methods

543 – 8 markers per chromosome was on the low side. 'Evenly spaced' here was presumably in physical distance since the marker spacing on the genetic map looks far from even. This also tells us something about the distribution of recombination along the chromosomes, which may be relevant to the cluster of divergent loci on Chr1 but this connection is not made in the manuscript.

620 – The rapid decline in LD suggests that window sizes for some analyses (e.g. 100kb for population genetic statistics) was rather large. This could have influenced sweep detection, for example.

627 – It is not clear why it was necessary to down-sample to 6 individuals per population here. Using 50kb windows are, again, rather large (and adjusting windows to a number of SNPs is preferable generally for this type of analysis), but I doubt either will have impacted the results strongly.

Figure S4 – Too many decimal places are used for both distance and Fst. The comma separator in the distance should be replaced with a point (or probably the fractions of km simply ignored).

Figure S7 – TreeMix also provides a plot of %variance explained, which helps to determine the appropriate number of migration events. This could usefully be given here.

Figure S21 – Are numbers above marker lines just their IDs? It might be helpful to indicate their linkage map positions (cf comments above about even spacing on the physical vs the genetic map).

*Reviewer #2 (Recommendations for the authors):*

Although I am enthusiastic about the promise of the system more generally and the importance of the questions posed, I have significant concerns about the manuscript and its conclusions. I am embedding my suggestions for new analyses with the descriptions of the issues above so that they are readable:

1) My primary concern in is in the design and interpretation of the QTL analysis. The QTL approach used here has low power, both due to the sample size and the number of markers used (it looks like ~8 per chromosome). The authors use an analysis of the sex determining locus as a "control" but because of the complete heritability of this trait in most systems it is more of a straw man to me. The authors conclude that the architecture of the trait is polygenic based on this, but we are missing key information to evaluate this. For example, what is the estimated heritability of the trait in this family and under laboratory conditions? I was also wondering if with the number of markers whether the authors might miss certain loci if the recombination rate is high.

2) There are some issues with the presentation and interpretation of the population genetic analyses. Many assumptions are made about whether introgression or ILS occurred and there are statements that are not accurate about it being "impossible" to distinguish between these scenarios. There are quite a lot of methods devoted to this problem, see e.g. (see Liu et al. PLoS Comp Bio 2014). Simpler tests like D-statistic and F4 ratio tests would add a lot of interpretability to these results.

3) Some of the analyses associated with ecological adaptation that follow on the QTL results struck me as ad hoc and with the potential to lead to spurious results. I am not familiar with the BayPass approach but since it is the approach that explicitly accounts for population structure it seems the one that would be most appropriate for the authors to focus on in a revised manuscript. The use of phylogenetic windows that associate with ecotype is concerning to me as given the level of ILS and gene flow that appears to be present in this system is would be very challenging to distinguish signal from noise. Related to this, in the companion manuscript the authors note that the chromosome 1 region has several inversions. I realize that the samples used in this manuscript are different but was unclear on how they excluded the possibility inversions were present on the chromosome 1 region that is highly differentiated in this analysis. This seems to be the primary region that distinguishes ecotypes in the BayPass analysis and thus is foundational to their conclusion that this signal is polygenic (since few other regions are significant in the BayPass analysis). I also noticed that there were not many QTL markers in this region of chromosome 1 regarding their previous analysis.

4) There were issues with the GO analysis that should be addressed. Because the gene universe used for GO enrichment is a subset of the full gene set, GO enrichment results will be biased. This will mostly lead to false positives (i.e. overrepresentation of a GO category due to evaluating a subset of genes that fall in that category). The authors should pair this with simulations or other analyses, or temper the strength of their claims.

---

## [Author Response]

Essential revisions:1) Describe the power of the QTL analysis and clearly describe its limitation. State that it remains unclear how complex the genetic basis of rhythmic versus arrhythmic behavior is, and that a moderately complex genetic basis (vulgo oligogenic) cannot be excluded.

We completely rewrote the section on QTL analysis. We now clearly describe its limitations (lines 203-229) and conclude that the phenotype is controlled by “more than one“ locus.

2) Be more open to the possibility that some of the apparent differentiation is due to demographic processes instead of adaptation only.

We have inserted a statement on the possible role of demographic processes in the Results section (lines 251-254), and also inserted a statement that part of the clustering of divergent alleles may be due to reduced recombination (lines 266-269).

3) Be more conservative with the GO enrichment analyses.

We maintain that we have been adequately cautious in the GO enrichment analysis. First, we conservatively only annotated orthologous genes with GO terms, avoiding false positives due to annotating duplicated genes (of essentially unknown function) with GO terms from their paralogs. Second, we think that the reviewer who is sceptic about our GO enrichment analysis has not fully appreciated the sophistication of our analysis. While details are given in the response below, briefly, we accounted for the GO term distribution of our reduced gene set when we specified the set of genes with GO terms as the custom background universe and only analyzed ecotype-associated genes with GO terms. The two subsets with GO terms are the same fraction of the respective full gene set, suggesting there are no biases in sub-setting. In our GO enrichment analysis, the Fisher’s exact test gives the exact hypergeometric probability under the null hypothesis for observing the degree of enrichment of a GO term given the custom gene universe, accounting for the potential differences in the representation of GO terms within the reduced gene set in comparison to the set of all genes. If there was any bias in the reduced gene universe, it would thereby have been accounted for in our analysis.

If the reviewer is concerned with the general stability of GO enrichment results between different subsets of a gene universe, we would like to highlight that Groß et al. have demonstrated that GO enrichment is surprisingly robust to changes in the GO term ontology and annotations (https://academic.oup.com/bioinformatics/article/28/20/2671/204694), further suggesting that using a reduced gene universe should not pose major problems.

Reviewer #1 (Recommendations for the authors):The Introduction should set the scene better before moving on to Clunio. Other cases of polygenic adaptation should be mentioned. There should be some discussion of theoretical expectations for the distribution of effect sizes for adaptive traits (with or without gene flow – see Yeaman and Whitlock 2011, for example).

We have re-written the introduction and now mention other cases of polygenic adaptation, as well as a discussion of the expected distribution of effect sizes for adaptive traits (lines 35-51).

The interpretations are generally strong and the core points are interesting and significant for the field – the recent, polygenic adaption from standing variation, the genomic clustering and the identification of candidate pathways are all strong results.

We thank the reviewer fort this positive evaluation of our work.

Evidence from the QTL analysis is least convincing – see specific comments below. The absence of evidence here is consistent with the results of the genome scan but that is about as far as one can go.

We have re-written this paragraph, and now conclude that we can exclude a single or few strong-effect loci. We also explicitly included a discussion of the weaknesses of the QTL data (lines 203-229). However, we still consider them as relevant support for the overall conclusions.

Absence of evidence for structural variants is also quite hard to interpret. I agree that the patterns do not obviously fit with a simple interpretation (such as a single polymorphic inversion) but there is remarkably strong LD over long distances. This might be explained by gene flow for the sympatric population pair but it appears to extend to other populations too. That said, I think the authors handle this well, leaving open the possibility that further work will reveal structural variants.

We agree that the evidence for SVs is hard to interpret. As the reviewer states we handled this well, we did not make any changes to the respective paragraphs.

Argument for standing variation does seem good – question about maintenance – plausibly suggest LA within the ancestral Atlantic ecotype.

We are happy the reviewer appreciates our argument for adaptation from standing genetic variation. We are not quite sure what she/he refers to with “LA”, but assume it is local adaptation. We make the point that standing variation for circadian timing is likely due to local adaptation in the discussion.

The GO enrichment results are very neat. A key strength is knowing the dominant axis of selection and being able to link that to genetic pathways likely to influence the relevant phenotypes (and so avoid 'story-telling'). These points are made in the Discussion but I think some of this could usefully be moved to a revised introduction, perhaps giving more emphasis to insights other than the polygenic nature of adaptation that can come from the unique features of this system.

We agree with the reviewer that the GO term enrichment results are neat, and hope that this helps to convince the other reviewer and the editor of the strength of the dataset (see detailed argumentation above and below). Indeed, knowing the dominant axes of selection is a particular strength in our system and we now mention this in the introduction (lines 92-94).

Below I give further comments by line number:15 – Not 'is' but 'can be' since speciation by other routes is possible and also because ecotype formation need not lead to speciation.

We have made the requested change.

36 – In addition to these recent references, it would be good to recognise long-standing awareness of the role of polygenic adaptation. See, for example, Le Corre and Kremer 2012 – doi: 10.1111/j.1365-294X.2012.05479.x.

We have added the respective citation.

38 – A polygenic basis can be demonstrated by quantitative genetics methods such as 'regional heritability'. It is currently unclear whether this is included under 'deep genomic and genetic analysis'.

Deep genomic and genetic analysis can also include Regional Heritability Mapping (RHM). However, as the method requires a pedigree and is thus not applicable to our data, we did not explicitly mention it here.

Figure 1 – The meaning of zero on the x-axis needs to be explained in the figure caption. The y-axis label (% emergence) seems to conflicts with the legend which says that these are plots of oviposition timings. Bergen is very far from the entrance to the Baltic – somewhere, the ecotype present in the Skagerrak needs to be mentioned.

There is no zero label on the x-axis. The zero label is on the y-axis (Emergence %).

The y-axis label does not conflict with the legend. The legend mentions “oviposition behavior” as one aspect in which the ecotypes differ. This was not listed in the figure, and may therefore have been misleading. We included “oviposition on exposed substrate” and “oviposition on water surface” in the figure. Otherwise, the legend clearly states that plots B-H are “lunar rhythms of adult emergence”.

Unfortunately, we have not sampled and we do not know any Clunio population from the Skagerrak. We have therefore included a comment that the wider overlap of the distribution Baltic and Arctic ecotypes must be subject to further investigation (lines 161-164).

118 – I am not sure why this is described as a contrast. The previous paragraph describes geographic rather than ecotype structure. That is surely confirmed by the observation that sympatric populations of different ecotypes are similar genetically.

The contrast referred to differentiated vs not differentiated populations. But this was indeed not clear from the context. So we removed the statement on this being a contrast (line 129).

154-164 – This inference should not be pushed too far since it is based on a single point in what is, presumably, a large area of overlap, perhaps with varying past and present mixes of the two ecotypes. So, the inferences apply to the specific sampled populations but may not generalise to the Atlantic-Baltic interaction at other locations. It also begs the question of whether this is a secondary contact, after an allopatric origin of the Baltic ecotype.

We agree with the reviewer that our conclusions are drawn based on a single point, and we believe we have stated this clearly that we are only referring to Bergen. In order to further clarify this, we have added the statement that further work will be required to see if there is a larger area of overlap, and if so which evolutionary scenarios (introgression, secondary contact) apply there (lines 161-164).

190 – How many topologies fit this pattern?

It is 357 topologies that fit the pattern. We have inserted the information in the text (line 200).

210 – Comparison to the SD locus only says that the number of loci for rhythmicity is likely to be >1, and even this conclusion is questionable since sex is a binary phenotype with no environmental variance. Therefore, to conclude from the lack of detectable QTL that 'many' loci underlie rhythmicity is going too far.

We agree with the reviewer and have rewritten the entire paragraph, delineating the shortcomings of the QTL mapping approach and concluding that the phenotype is not controlled by a single large-effect locus.

It may be possible to use simulation to get a better idea of the power of the QTL analysis and so the likely maximum effect size.

While it might be possible to estimate the maximum effect size from simulations, we think that this effort goes beyond the scope of our manuscript and would actually be distracting. We have instead followed the editor’s suggestion and have clearly spelled out the limitations of the QTL analysis and have tempered our claims to say the trait is controlled by “more than one” locus (lines 203-229).

If the vertical lines in Figure 4 represent marker loci (not specified in the legend) then the number is rather low and distribution rather uneven, perhaps making QTL detection in some parts of the genome difficult (including the important region in the centre of Chr1).

The vertical lines in Figure 4 represent marker loci, and we have added this information to the figure legend. We have also added the statement that the low marker number limits QTL mapping in the respective paragraph (lines 219-220). The uneven distribution of markers may make QTL detection difficult, but not for the important region in the center of Chr1 one, which is covered by the markers with low recombination at the left end of the linkage map. This in turn is important in the light of the reviewers comment below on lines 543-8, and we have therefore added this information to the manuscript (lines 264-269) and the “Figure 4—figure supplement 2”.

I also wonder how good a measure of rhythmicity is provided by the emergence day (as an absolute deviate, if I understand correctly) for an individual. This seems like a difficult trait to handle since rhythmicity cannot easily be measured for any one individual.

Rhythmicity is indeed a difficult phenotype and we have now clearly stated its specific problems in both the methods (lines 548-550) and the Results sections (lines 219-224). We can safely assume that individuals emerging outside the peak are arrhythmic. However, arrhythmic individuals which happen to emerge during the peak are scored as rhythmic, leading to a fraction of individuals with false phenotypes. We have come up with various phenotyping procedures, which are not reported here. This included the exclusion of days at the edges of the peak, which are particularly prone to phenotyping error. It also included an expectation-maximization (EM) algorithm that would assign phenotypes to the individuals in the peak based on the probability of being rhythmic and in a way so that the LOD score of the resulting phenotype panel, i.e. the probability of finding a significant QTL, is maximized. This algorithm successfully allowed QTL mapping based on rhythmicity in another case – see our preprint at https://doi.org/10.1101/2022.10.12.511720. In this preprint, the different scoring schemes and the EM algorithm converged to 1-3 detectable QTL for rhythmicity, notably based on smaller mapping families (but more markers) than used here. For the cross reported here, no scoring scheme led to detectable QTL and the EM algorithm did not converge to reproducible QTL. This may underscore that the genetic basis of arrhythmicity in the case we report here may be based on more than a few loci. But it may in part also be attributable to the difficult phenotype. Hence, in order not to over-analyze and over-interpret, we decided to only report the simplest phenotyping scheme and considerably temper our claims to say “more than one locus” contributes to the arrhythmic phenotype.

280 – It is surprising that the cluster of outliers on Chr1 does not appear as a QTL, especially given the long-range LD in this area

It is surprising that the region does not show up as a QTL. This may underscore that other loci are likely to contribute.

Methods543 – 8 markers per chromosome was on the low side. 'Evenly spaced' here was presumably in physical distance since the marker spacing on the genetic map looks far from even. This also tells us something about the distribution of recombination along the chromosomes, which may be relevant to the cluster of divergent loci on Chr1 but this connection is not made in the manuscript.

We have added the information that ‘evenly spaced’ is with respect to physical distance (lines 531-533). We have also revised “Figure 4—figure supplement 2” (previously Supplementary Figure 21) to include the comparison between physical distance and genetic map distance. We have included the information that the divergent region on chr1 has low recombination, which may aid the clustering of divergent loci (lines 266-269).

620 – The rapid decline in LD suggests that window sizes for some analyses (e.g. 100kb for population genetic statistics) was rather large. This could have influenced sweep detection, for example.

In this case our methods section was unfortunately incomplete. Figure 6 (sweep detection) reports 10 kb windows with 1kb steps. We have added the information to the methods (lines 613-615) and the figure legend. Additionally, we have tried 1kb windows for LD and 5 kb windows for pi, but these plots were very noisy, likely due to the limited number of SNPs in such small windows.

627 – It is not clear why it was necessary to down-sample to 6 individuals per population here. Using 50kb windows are, again, rather large (and adjusting windows to a number of SNPs is preferable generally for this type of analysis), but I doubt either will have impacted the results strongly.

We included the explanation that running-times of the topology weighting algorithm increases exponentially with the number of individuals in the methods section (lines 628-632). With six individuals it is 6^6^ = 46,656 combinations, with the full dataset it would have been 6^24^ = 4.7*10^18^ combinations. That is why we had to down-sample.

While 50kb windows are indeed large, we also tried 5kb windows and did not observe any notable differences. See Author response image 1.

**Author response image 1. sa2fig1:** 

Figure S4 – Too many decimal places are used for both distance and Fst. The comma separator in the distance should be replaced with a point (or probably the fractions of km simply ignored).

We have adjusted the figure accordingly.

Figure S7 – TreeMix also provides a plot of %variance explained, which helps to determine the appropriate number of migration events. This could usefully be given here.

We couldn’t find the respective plotting function in TreeMix. Instead we prepared a plot of the model likelihood for zero to five migration events. The sharp flattening after one migration event suggests one migration event produces the only significant improvement of the model (new “Figure 2—figure supplement 7”).

Figure S21 – Are numbers above marker lines just their IDs? It might be helpful to indicate their linkage map positions (cf comments above about even spacing on the physical vs the genetic map).

Yes, the numbers above markers are their IDs. We have adjusted the figure to show the position of the markers both on the linkage map and the physical map, as well as relative to peaks in F_ST_ (now “Figure 4—figure supplement 2”). This also allowed us to insert statements about reduced recombination in certain areas of the genome (lines 266-269).

Reviewer #2 (Recommendations for the authors):Although I am enthusiastic about the promise of the system more generally and the importance of the questions posed, I have significant concerns about the manuscript and its conclusions. I am embedding my suggestions for new analyses with the descriptions of the issues above so that they are readable:1) My primary concern in is in the design and interpretation of the QTL analysis. The QTL approach used here has low power, both due to the sample size and the number of markers used (it looks like ~8 per chromosome). The authors use an analysis of the sex determining locus as a "control" but because of the complete heritability of this trait in most systems it is more of a straw man to me. The authors conclude that the architecture of the trait is polygenic based on this, but we are missing key information to evaluate this. For example, what is the estimated heritability of the trait in this family and under laboratory conditions? I was also wondering if with the number of markers whether the authors might miss certain loci if the recombination rate is high.

We agree that our QTL analysis has limitations in terms of sample size, as well as number and distribution of markers. To fully acknowledge these limitations, we have now spelled them out in the results and methods section. We have also tempered our claims significantly, and now only state that QTL analysis suggests that the trait is controlled by more than one gene (lines 203-229). We have not included an estimate of heritability, as it is not straightforward to calculate it with the data we have at hand. Also, we do not over-emphasize the results of the QTL analysis – we rather acknowledge that it has limitations. Still, its results complement the whole analysis.

2) There are some issues with the presentation and interpretation of the population genetic analyses. Many assumptions are made about whether introgression or ILS occurred and there are statements that are not accurate about it being "impossible" to distinguish between these scenarios. There are quite a lot of methods devoted to this problem, see e.g. (see Liu et al. PLoS Comp Bio 2014). Simpler tests like D-statistic and F4 ratio tests would add a lot of interpretability to these results.

We added the f4 ratio test to our analysis and found that the genome-wide f4 ratio is significantly different from 0, i.e. there is some introgression. However, the signature of introgression is driven by only 775 SNPs, a small fraction of the total dataset. We have added this information in the manuscript and re-phrased our interpretation accordingly (lines 187-196).

3) Some of the analyses associated with ecological adaptation that follow on the QTL results struck me as ad hoc and with the potential to lead to spurious results.

It would have been helpful if the reviewer had explicitly stated which analyses struck her/him as ad hoc. We assume it is the analyses referred to below.

I am not familiar with the BayPass approach but since it is the approach that explicitly accounts for population structure it seems the one that would be most appropriate for the authors to focus on in a revised manuscript. The use of phylogenetic windows that associate with ecotype is concerning to me as given the level of ILS and gene flow that appears to be present in this system is would be very challenging to distinguish signal from noise.

We present three different approaches to detect genomic regions associated with ecotype: phylogenetic windows, BayPass and F_ST_ outliers. One of the strengths of our analysis is that the three approaches lead to very similar results, e.g. the clustering of ecotype-associated variants on chr1. Hence, we would not like to drop any of the analyses.

We agree with the reviewer that BayPass is indeed the strongest approach. That is why in the manuscript we already focus on the BayPass results: After reporting on all three approaches (phylogenetic windows, BayPass and F_ST_ outliers) in one paragraph each, the entirety of the downstream analyses (adaptation from standing variation, GO term analyses, highly associated loci) is based solely on the BayPass results.

While the phylogenetic approach has its weaknesses due to ILS and gene flow, we would still like to keep the single sentence reporting on it (lines 235-236), as the results are fully consistent with the other analyses.

Related to this, in the companion manuscript the authors note that the chromosome 1 region has several inversions. I realize that the samples used in this manuscript are different but was unclear on how they excluded the possibility inversions were present on the chromosome 1 region that is highly differentiated in this analysis. This seems to be the primary region that distinguishes ecotypes in the BayPass analysis and thus is foundational to their conclusion that this signal is polygenic (since few other regions are significant in the BayPass analysis). I also noticed that there were not many QTL markers in this region of chromosome 1 regarding their previous analysis.

We also report on putative inversions on chromosome 1 in this manuscript, represented by blocks of observed long-range LD (lines 264-266; “Figure 3—figure supplement 8”, previously Supplementary Figure 17). These putative inversions overlap with the divergent region on chromosome 1 and reduce recombination in that region, as can be seen from long-range LD, as well as the combination of physical markers distances with map length in our revised “Figure 4—figure supplement 2” (previously Supplementary Figure 21). However, we cannot find a 1:1 association of these putative inversions with the divergent region (the boundaries differ markedly), nor with ecotype (putative inversions are present in both ecotypes and rather population-specific than ecotype-specific). In a PCA of the divergent region, the individuals do not segregate in any pattern that would be typical for inversion haplotyopes (as is observed in the companion manuscript).

We would also like to point out that it is not the case that few other regions outside the chromosome 1 cluster are significant in BayPass. In our figure 3 the red dots are only the extreme outliers (p < 10^-10^), but black dots are already significant at p < 0.001 (as is stated in the figure legend). Thus, ecotype associated SNPs are all over the genome, and this is foundational to the conclusion that the signal is polygenic.

The divergent region on chromosome 1 is actually covered by four out of eight markers on that chromosome. This may not have been clear in the previous version of our manuscript, and we have therefore revised “Figure 4—figure supplement 2” (previously Supplementary Figure 21) to make the connection between the linkage map and physical marker locations.

4) There were issues with the GO analysis that should be addressed. Because the gene universe used for GO enrichment is a subset of the full gene set, GO enrichment results will be biased. This will mostly lead to false positives (i.e. overrepresentation of a GO category due to evaluating a subset of genes that fall in that category). The authors should pair this with simulations or other analyses, or temper the strength of their claims.

We see that our conservative approach to annotate only the subset of orthologous genes with GO terms produces an issue, namely that the genes without GO terms will be missing from the analysis, i.e. there are false negatives. In order to address this problem we have performed a second set of analysis which takes into account all genes with highly divergent SNPs irrespective of whether or not they are annotated with GO terms (lines 350-388). While we found that indeed many divergent genes are not annotated with GO terms, the functions we can assign to these genes by investigation of the relevant literature match the results from GO term analysis very well. This is the best we can do about false negatives.

The reviewer suggests there might be false positives in our analysis. We disagree with that for several reasons:

First, we think there might be a misunderstanding of the analysis due to insufficient information provided in the methods and we apologize for it. In our topGO analysis, the reduced dataset of 5,393 genes with GO terms was specified as the custom background gene universe. By specifying this reduced gene universe to topGO, with the parameter ‘allGenes’, any potential differences in the background frequencies of certain GO terms between our reduced gene set and the set of all genes will be statistically accounted for by the tool, through incorporation into the null hypothesis. 463 out of 1,400 ecotype-associated gene models were annotated with GO terms and only those were fed into the topGO analysis. The p-value returned by the Fisher’s exact test implemented in topGO calculates exactly the probability of observing at least the same amount of enrichment of a GO term when 463 genes are randomly selected out of the specified gene universe of 5,393 genes. Thus, if there is any bias in GO terms of our custom gene universe, this should have been accounted for. We also note that the fraction of genes with GO terms among the ecotype-associated genes and among the total genes is not significantly different (chi-squared test, p=0.12), indicating that both sets of genes are affected in the same way by the lack of GO term annotations.

Second, the reviewer asks for simulations to support our claims. If the reviewer is concerned with the ‘Open World Assumption’ that is inherent in Gene Ontology, namely that we have made statements about whether or not enrichment is true without knowing all observations (i.e. all GO terms for all genes) and would like us to perform simulations to evaluate the impact of using varying subsets of the gene universe on the stability of enrichment results -- Here we can say that Groß et al. have investigated the difference in results of functional analyses using GO annotations in both primates and rodents from different GO versions (https://academic.oup.com/bioinformatics/article/28/20/2671/204694), i.e. different Gene universes and levels of completeness. Their conclusions are that enrichment results are surprisingly stable to changes in ontology and annotation. Hence, we conclude that our reduced gene universe will very likely not cause major issues to the reliability of the conducted enrichment analysis.

In order to avoid these possible misunderstandings, we have inserted the respective information in the methods section (lines 670-681).

We also want to point out that the other reviewer considers our GO term enrichment results particularly strong.